# A novel human fetal lung-derived alveolar organoid model reveals mechanisms of surfactant protein C maturation relevant to interstitial lung disease

Kyungtae Lim [1,2,3,17], Eimear N Rutherford [4,17], Livia Delpiano [1,2], Peng He [5,6], Weimin Lin [1], Dawei Sun [1,2,7], Dick J H Van den Boomen [8,9], James R Edgar [10], Jae Hak Bang [11], Alexander Predeus [5], Sarah A Teichmann [5,12], John C Marioni [5,6,13,16], Lydia E Matesic [14], Joo-Hyeon Lee [2,11], Paul J Lehner [8], Stefan J Marciniak [4,15], Emma L Rawlins [1,2 ✉] & Jennifer A Dickens [4,15 ✉]

## Abstract

**Alveolar type 2 (AT2) cells maintain lung health by acting as stem cells and producing pulmonary surfactant. AT2 dysfunction underlies many lung diseases, including interstitial lung disease (ILD), in which some inherited forms result from the mislocalization of surfactant protein C (SFTPC) variants. Lung disease modeling and dissection of the underlying mechanisms remain challenging due to complexities in deriving and maintaining human AT2 cells ex vivo. Here, we describe the development of mature, expandable AT2 organoids derived from human fetal lungs which are phenotypically stable, can differentiate into AT1-like cells, and are genetically manipulable. We use these organoids to test key effectors of SFTPC maturation identified in a forward genetic screen including the E3 ligase ITCH, demonstrating that their depletion phenocopies the pathological SFTPC redistribution seen for the SFTPC-I73T variant. In summary, we demonstrate the development of a novel alveolar organoid model and use it to identify effectors of SFTPC maturation necessary for AT2 health.**

**Keywords** Stem Cell; Pulmonary Fibrosis; Surfactant Protein C; E3 Ligase; ITCH
**Subject Categories** Development; Methods & Resources

## Introduction

Despite its role in increasing the stability of pulmonary surfactant, SFTPC is not absolutely required for lung development and surfactant secretion (Glasser et al, 2001). However, its aberrant handling during intracellular trafficking and maturation results in toxic gain-of-function effects. This is demonstrated by the pathogenic variant I73T, which accumulates immature isoforms at the plasma membrane and causes heritable forms of pulmonary fibrosis (Brasch et al, 2004; Alysandratos et al, 2021). Study of this variant in immortalized cells has suggested SFTPC trafficking into multivesicular bodies (MVBs) is indirect (via the plasma membrane) and requires its ubiquitination (Dickens et al, 2022). Despite the substituted amino acid (I73) lying on the opposite side of the membrane to the ubiquitinated site at K6, failure of ubiquitination appears to be the key determinant of SFTPC-I73T redistribution and the cause of Alveolar Type 2 (AT2) cell dysfunction and consequently ILD. Immortalized cells can be used to generate hypotheses, but questions regarding mechanisms and key effectors of SFTPC trafficking can ultimately only be answered using genetically manipulable physiological AT2 cells which endogenously process surfactant.

AT2 cells act as stem cells for the alveolar epithelium (Desai et al, 2014; Barkauskas et al, 2013). AT2 organoids have been grown from human adult lungs and used as models of SARS-CoV-2 infection (Youk et al, 2020; Katsura et al, 2020; Salahudeen et al, 2020; Ebisudani et al, 2021; Chiu et al, 2022; Lamers et al, 2021) and mixed populations of adult lung progenitors have been expanded as

[1]Wellcome Trust/CRUK Gurdon Institute, University of Cambridge, Cambridge CB2 1QN, UK. [2]Department of Physiology, Development and Neuroscience, University of Cambridge, Cambridge CB2 3DY, UK. [3]Department of Life Sciences, Korea University, 145 Anam-Ro, Seoungbuk-Gu, Seoul 02841, South Korea. [4]Cambridge Institute for Medical Research, Cambridge CB2 0XY, UK. [5]Wellcome Sanger Institute, Wellcome Genome Campus, Hinxton, Cambridge CB10 1SA, UK. [6]European Molecular Biology Laboratory, European Bioinformatics Institute (EMBL-EBI), Wellcome Genome Campus, Hinxton, UK. [7]Broad Institute of Massachusetts Institute of Technology and Harvard, Cambridge, MA 02142, USA. [8]Cambridge Institute of Therapeutic Immunology and Infectious Disease, Jeffrey Cheah Biomedical Centre, University of Cambridge, Cambridge CB2 0AW, UK. [9]Harvard Medical School, Department of Cell Biology, Harvard University, LHRRB building, 45 Shattuck Street, Boston, MA 02115, USA. [10]Department of Pathology, University of Cambridge, Cambridge CB2 1QP, UK. [11]Wellcome-MRC Cambridge Stem Cell Institute, Jeffrey Cheah Biomedical Centre, University of Cambridge, Puddicombe Way, Cambridge CB2 0AW, UK. [12]Theory of Condensed Matter Group, Department of Physics, Cavendish Laboratory, University of Cambridge, Cambridge, UK. [13]Cancer Research UK Cambridge Institute, University of Cambridge, Robinson Way, Cambridge CB2 0RE, UK. [14]Department of Biological Sciences, University of South Carolina,, 715 Sumter St., Columbia, SC 29208, USA. [15]Royal Papworth Hospital, Papworth Road, Trumpington CB2 0AY, UK. [16]Present address: Genentech, South San Francisco, CA, USA. [17]These authors contributed equally as first authors: Kyungtae Lim, Eimear N Rutherford. ✉E-mail: elr21@cam.ac.uk; jac72@cam.ac.uk

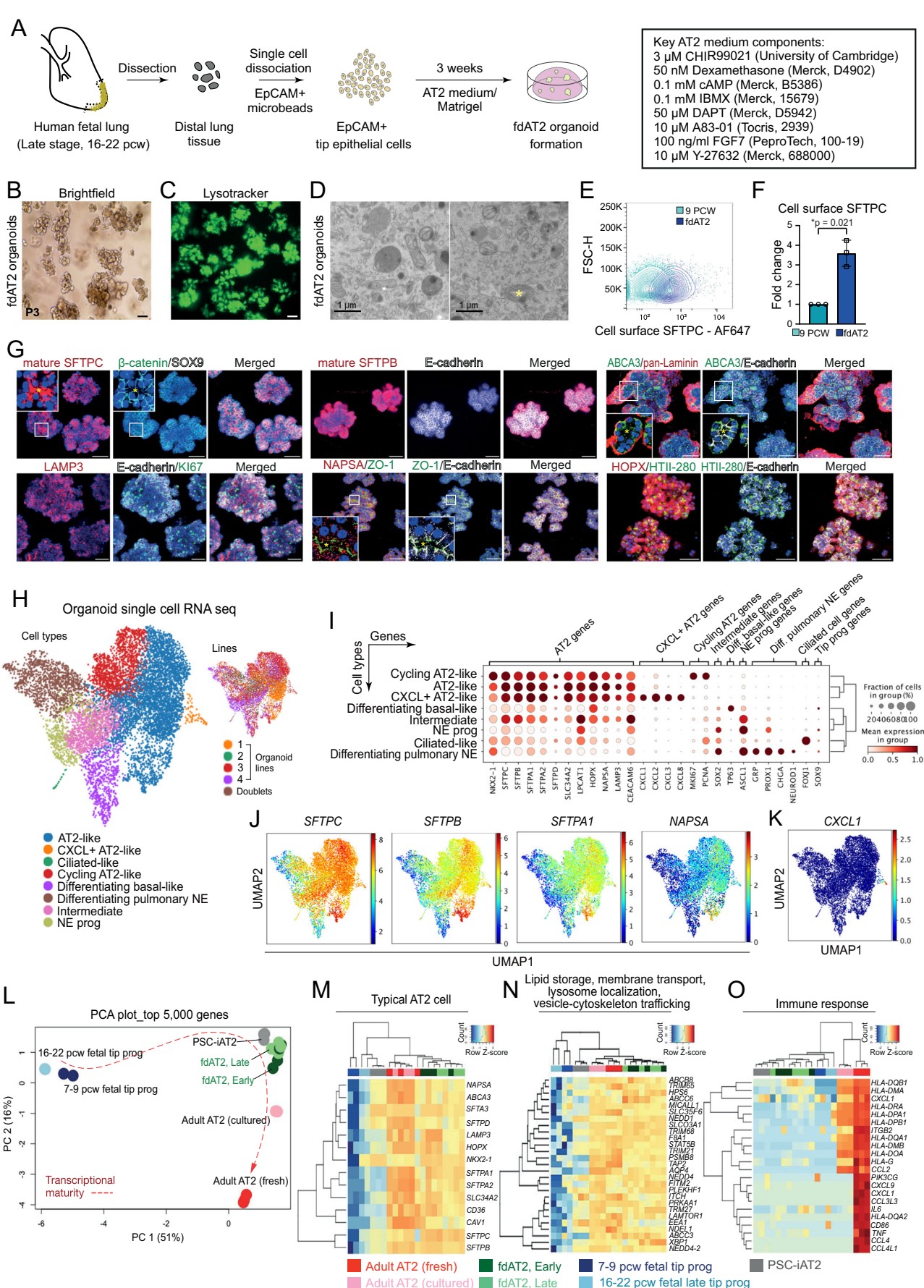

◀ **Figure 1.   Human fetal-derived alveolar type 2 organoids show similar surfactant protein production, trafficking, and secretion to adult AT2 cells.**

(A) Experimental scheme and culture medium for derivation and establishment of fdAT2 organoids from human fetal lungs at 16–22 pcw. (B) Bright-field images of fdAT2 organoids at P3 which were established from 19 pcw fetal lung tissue. Scale bar, 50 μm. (C) Uptake of lysotracker green DND-26 that stains acidic compartments within an fdAT2 organoid line at P14, showing the accumulation of lamellar bodies (acidic lysosome-related organelles). Scale bar, 50 μm. (D) Electron microscopy showing the presence of lamellar bodies with characteristic concentric lamellar membranes within the cytosol (left panel) and lying between two cells following exocytosis (*, right panel). Scale bar, 1 μm. (E) Flow cytometric analysis of cell surface proSFTPC in 9 pcw lung tip progenitor organoids and fdAT2 organoids as measured by C-terminal SFTPC antibody which recognizes the full-length protein. Quantified in (F), mean ± SD, $n = 3$ biologically independent lines of each organoid type (paired two-tailed Student's $t$ test). (G) Immunofluorescence images of fdAT2 organoids using surfactant protein production-associated markers including mature SFTPC and SFTPB, LAMP3, NAPSA, and ABCA3, plus typical alveolar type 2 cell lineage markers HTII-280 and HOPX, epithelial cell polarity markers E-cadherin, pan-Laminin, and ZO-1, and a proliferation marker, KI67. *Asterisk (yellow) indicates apical lumen. Organoids were at P10-15. DAPI (blue), nuclei. Images are shown as confocal z projections. Scale bar, 50 μm. (H) UMAP visualization of 9619 cells from 4 organoid lines at P11–16, colored by cell type (left), organoid line (right). (I) Dot plot describing differential marker gene expression. (J, K) UMAP plots showing transcript expression of AT2 lineage markers *SFTPC*, *SFTPB*, *SFTPA1*, *NASPA* (J) and *CXCL1* (K). (L) Principal component analysis (PCA) plot of transcriptomic profiles of the fdAT2 organoids at early and late passages (fdAT2; early and late), lung tip progenitor organoids from 7–9 pcw and 16–22 pcw, and other alveolar type 2 cells that were previously reported, PSC-iAT2, adult AT2 cells cultured or freshly isolated from adult human lung Abo et al, 2022; Jacob et al, 2017. (M–O) Heatmap of DEGs associated with typical AT2 cell fate (M), trafficking (N), and immune response (O). A list of trafficking-related genes was selectively obtained following gene ontology (GO) analysis for the GO terms; lipid storage, membrane transport, lysosome localization, vesicle-cytoskeleton trafficking; see also Figs. EV1–3, Movie EV1 and Appendix Fig. S1. Source data are available online for this figure.

organoids and differentiated to AT2 cells (Chiu et al, 2022). However, the typical approach is to expand human adult AT2 organoids which proliferate slowly and are difficult to genetically manipulate. AT2 organoids can also be derived from pluripotent stem cells (PSC-iAT2s) (Jacob et al, 2017; Abo et al, 2022; McCauley et al, 2017). PSC-iAT2s can readily be genetically manipulated, expanded, and efficiently differentiated to AT1 cells, but their differentiation is complicated (Jacob et al, 2019) and PSC-iAT2s can spontaneously dedifferentiate to other organ lineages (McCauley et al, 2017). A complementary method for growing genetically manipulable human AT2 cells would be of great value for investigating surfactant trafficking and lung disease.

During human lung development, the distal tip epithelial cells act as multipotent progenitors (Nikolić et al, 2017; Miller et al, 2020). From ~15 pcw (postconception weeks) the human tips retain progenitor marker expression, but also upregulate markers of AT2 cells and immature AT2 cells appear in the tissue from 17 pcw (Lim et al, 2023; He et al, 2022). We have recently shown that 16–22 pcw epithelial tip cells can be expanded as organoids and differentiated to AT2 cells (Lim et al, 2023). However, these previous fetal-derived AT2 organoids were not proliferative, limiting their use for functional studies and genetic manipulation. We have now developed a highly robust, efficient, and scalable culture condition that induces the differentiation of 16–22 pcw fetal lung tip cells into mature AT2 cells which grow as expandable 3D organoids. Here, we characterize the fetal-derived AT2 organoids (hereafter fdAT2), showing that they are stable over long-term passaging, efficiently process and secrete surfactant, and can differentiate into AT1-like cells in vitro and in mouse lung transplantation assays. We use a forward genetic screen to identify candidate effectors of SFTPC trafficking which we validate using CRISPR interference (CRISPRi) in the fdAT2 organoids. We demonstrate that trafficking of SFTPC requires ubiquitination by the HECT domain E3 ligase ITCH, and that its depletion phenocopies the redistribution seen for the pathological SFTPC$^{I73T}$ variant.

## Results

Our AT2 medium directly induces the differentiation of 16–22 pcw fetal lung tip cells into mature AT2 cells which grow as expandable

3D organoids (Fig. 1A–D). The fdAT2 organoids are derived from the distal tip epithelium and/or immature AT2 cells, and not from the more proximal SCGB3A2$^+$ airway progenitors which do not efficiently form organoids under these conditions (Fig. EV1A–C). The fdAT2 organoids form within 3 weeks and can be split every week, for at least 20 passages, while sustaining *SFTPC* promoter-GFP reporter and AT2 marker expression (Fig. EV1D,E; Appendix Fig. S1). They retain these characteristics following cryopreservation and thawing.

The fdAT2 organoids show mature AT2 cell features, including production and secretion of mature forms of SFTPB, SFTPC and lamellar bodies (Figs. 1D,G and EV1F–H; Movie EV1). Moreover, the organoids express proteins required for surfactant production, such as LAMP3, ABCA3, and Napsin-A, as well as typical AT2 markers HTII-280, E-cadherin and HOPX (Travaglini et al, 2020), whereas a fetal tip progenitor marker, SOX9 was not detected (Fig. 1G; Appendix Fig. S1). There is some variability in protein levels of these markers in the organoids (illustrated clearly in Fig. EV1H and Appendix Fig. S1G). Immature SFTPC is detectable at the plasma membrane in fdAT2 organoid cells, supporting our previous model of SFTPC trafficking (Fig. 3C) in which proprotein transits the cell surface before it is endocytosed and cleaved en route to later compartments (Dickens et al, 2022) (Fig. 1E,F). Proliferation of the fdAT2 organoids was greatly reduced, and expression of genes associated with surfactant trafficking increased, when FGF7 was removed from the medium (Fig. EV1I–K), suggesting FGF7 is an AT2 mitogen, consistent with mouse data (Brownfield et al, 2022), and that cell cycle exit further increases surfactant processing. These culture conditions induce the differentiation of 16–22 pcw lung epithelial progenitors into mature AT2 cells which maintain identity and function during prolonged passaging.

To further characterize the fdAT2 organoids, we performed single-cell RNA sequencing (scRNA-seq) of four independent lines at passage 11–16 (Figs. 1H and EV2A). When visualized as a uniform manifold approximation and projection (UMAP) these data confirmed the majority of organoid cells are AT2-like, including proliferative AT2s and a small sub-population co-expressing chemokine genes (Fig. 1H–K). In addition, differentiating basal and neuroendocrine cells were present (Figs. 1H,I and EV2). These were immature, non-proliferative cells, largely

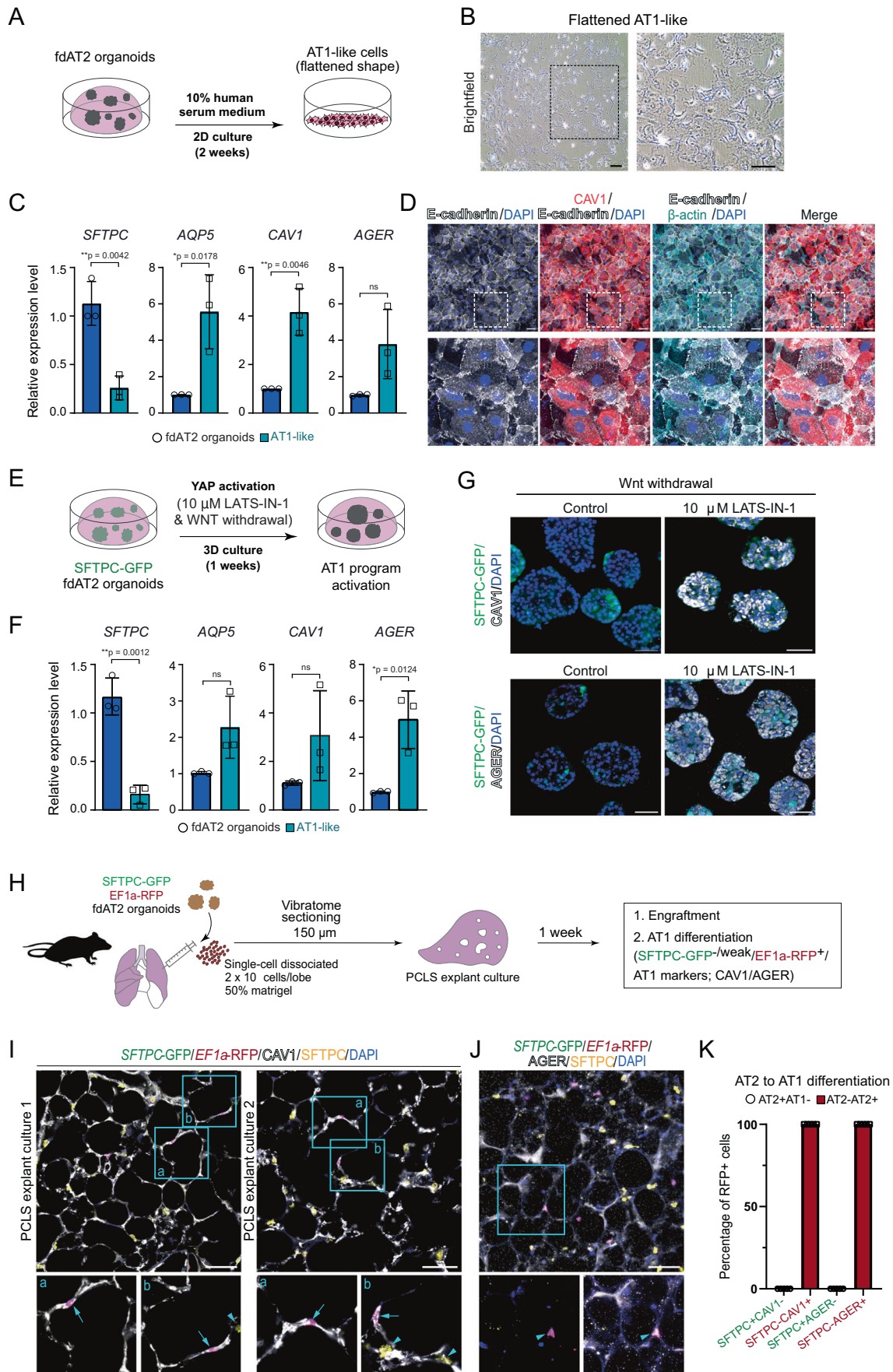

**Figure 2.   Alveolar type 1 cell fate differentiation of AT2 organoids under in vitro and ex vivo culture conditions.**

(A–D) Alveolar type 1 cell (AT1) fate differentiation of fdAT2 organoids in 10% human serum-containing medium on 2D culture. Experimental scheme of AT1 differentiation (A), bright-field image of AT2 organoids upon AT1 differentiation (B), RT-qPCR analysis of AT1 markers *AQP5*, *CAV1*, and *AGER*, and an AT2 marker, *SFTPC* (C), and immunofluorescence imaging with CAV1, E-cadherin, β-actin, and DAPI (nuclei); scale bar, 50 μm (D). For RT-qPCR, data were normalized to fdAT2 organoids in AT2 medium; mean ± SD, *n* = 3 biologically independent organoid lines (unpaired *t* test (two-tailed)). (E–G) AT1 differentiation of *SFTPC*-GFP-expressing fdAT2 organoids in a culture medium containing YAP signaling agonist, 10 μM LATS-IN-1, in the absence of Wnt agonists, in 3D organoid culture. Experimental scheme of AT1 differentiation (E), RT-qPCR analysis of AT1 markers *AQP5*, *CAV1*, and *AGER*, and an AT2 marker, *SFTPC* (F), and immunofluorescence imaging with CAV1, AGER, and DAPI (nuclei); scale bar, 50 μm (G). For RT-qPCR, data were normalized to fdAT2 organoids in AT2 medium; mean ± SD, *n* = 3 biologically independent organoid lines (unpaired *t* test (two-tailed)). (H) Explant cultured mouse precision-cut lung slices (PCLS) 1 week post-injection of *SFTPC*-GFP; *EF1a*-RFP human fdAT2 organoids. *n* = 3 biologically independent lines of fdAT2 organoids were used for the explant culture (16402, 16587, 16392). (I, J) Immunofluorescence images of PCLS showing engrafted and AT1 differentiated human cells. RFP[+] cells were monitored by a combination of cell type marker antibodies against SFTPC (I, J), and CAV1 (I) or AGER (J). Arrows, AT1 differentiated human cells. Arrowheads, mouse AT2 cells. DAPI, nuclei. Scale bar, 50 μm. Quantitation (K) of AT1 lineage positive human cells in the explants, by measuring the proportion of AT2 (SFTPC) and AT1 markers (CAV1 and AGER) co-localizing with RFP (mean ± SD; *n* = 53, SFTPC/CAV1 cells; *n* = 56, SFTPC/AGER cells; 6 lung slices from 3 biological replicates). See also Movies EV2–4. Source data are available online for this figure.

co-expressing *SFTPC* and other canonical AT2 markers, suggesting they originate from AT2-like cells, consistent with recent reports that hypoxia induces airway differentiation of AT2 cells (Scott McCall et al, 2023; Dong et al, 2024). Integration of fdAT2 organoid scRNA-seq with published human adult (Madissoon et al, 2023) and fetal (He et al, 2022) lung cell atlases broadly confirmed our cell type annotations (Fig. EV2L–P).

We also performed bulk RNA-seq to compare organoid stability over repeated passaging, comparing the transcriptome of the four lines at early passage (fdAT2 early; P1, 4, 6, 7) and late passage (fdAT2 late; P12, 13, 16, 17). This allowed us to also compare pluripotent stem cell-derived 3D cultured induced AT2 (PSC-iAT2) (Jacob et al, 2017), freshly isolated and 2D cultured adult AT2 cells (Abo et al, 2022) and 8–9 pcw and 16–20 pcw fetal lung tip progenitor organoids (Lim et al, 2023) (Fig. 1L). Our fdAT2 organoids clustered with other AT2 cells but were distinct from the fetal lung tip progenitors (PC1, Figs. 1L and EV3A). The cultured adult AT2 cells sat between the PSC-iAT2 and fdAT2 organoids, and the freshly isolated adult AT2 cells, indicating a culture-effect on the gene expression profile of adult AT2 cells (Alysandratos et al, 2022) (Fig. 1L). A direct comparison of the fdAT2s and adult cultured AT2s revealed that the most significantly increased transcripts in fdAT2s are related to cell division (Fig. EV3B), consistent with their expandability. Importantly, the transcriptional profile of the fdAT2 organoids remained stable up to passage 17 in this experiment, suggesting that expansion of these organoids does not affect AT2 cell identity/function (Figs. 1L and EV3A,E).

To further characterize the fdAT2 organoids, we extracted AT2-specific differentially expressed genes (DEGs) by comparing AT2 cells from each source to 16–20 pcw fetal tip progenitor organoids (Fig. EV3C; log$_2$FC > 2, *P* value < 0.05; Datasets EV1 and 2). Comparison of DEGs identified gene expression profiles that distinguish the different AT2 cells. 51% of DEGs (4083) were commonly shared across all AT2 cell sources (Fig. EV3C; center). Gene ontology (GO) analysis showed that these genes are related to cytoplasmic translation, protein transport, vesicle-mediated transport, and ER–Golgi vesicle-mediated transport (Fig. EV3D). Consistent with this, the fdAT2 organoids exhibited a gene expression profile related to surfactant protein synthesis (Figs. 1M and EV3E). Our DEG analysis also identified subsets of genes that were partially shared, or uniquely expressed, in the different AT2 cells (Fig. EV3C). For example, GO analysis for 532 DEGs that are shared by fdAT2 organoids and adult AT2 cells, but not by PSC-

iAT2, showed terms associated with vesicle cytoskeletal trafficking, lipid storage, transmembrane transport, and lysosome localization (Figs. 1N and EV3D',E). These GO terms are strongly correlated with the physiological surfactant-producing function of the AT2 cells (Fig. 1C–G), highlighting the utility of the fdAT2 organoids to study surfactant processing. However, the fetal-derived and iPSC-derived AT2 organoids were missing 355 DEGs related to antigen processing and presentation via MHC class II and immune response that only the adult AT2 cells expressed (Figs. 1O and EV3D",F), suggesting that immune function cannot be acquired in a cell-autonomous manner in vitro.

Next, we investigated how far the transcriptional state of the fdAT2 organoids resembles the PSC-iAT2. Their transcriptome is broadly similar (Figs. 1L and EV3G; Dataset EV3). However, gene set enrichment analysis (GSEA) of >2000 DEGs showed that the fdAT2 organoids were enriched for genes associated with surfactant metabolism compared to PSC-iAT2 (Fig. EV3G–J), although these differences likely reflect variation in culture conditions. Overall, these data confirm that fdAT2 organoids are highly proliferative, but also strongly resemble adult AT2 cells at a transcriptional level and possess the capacity for mature surfactant protein production, trafficking, and secretion. However, the fdAT2 organoids lack immune response-related features and are not suitable for studying this aspect of AT2 cells. This may reflect some immaturity, or, more likely, the sterile environment in which they are grown; supported by the observation that adult AT2 gradually lose their immune signature when cultured (Figs. 1O and EV3D").

Adult AT2 cells function as facultative progenitors of lung alveoli that can self-renew and differentiate into alveolar type 1 (AT1) cells to replenish AT1 cells upon injury (Desai et al, 2014; Barkauskas et al, 2013). Our AT2 organoid medium contains Wnt agonists which inhibit AT1 cell differentiation (Burgess et al, 2024). We investigated whether the fdAT2 organoids have the capacity to differentiate into AT1-like cells. The fdAT2 organoids were treated with AT1 lineage-promoting conditions: 10% human serum medium on 2D for 2 weeks (Katsura et al, 2020), or 10 μM LATS-IN-1, an inhibitor of LATS1/2 kinases causing activated YAP signaling, in 3D for 1 week (Burgess et al, 2024) (Fig. 2A–G). In both conditions, the fdAT2 organoids showed upregulation of AT1 fate markers, such as *AQP5*, *CAV1*, and *AGER*, and downregulation of *SFTPC* and *SFTPC*-GFP (Fig. 2C,D,F,G). Next, we tested whether the fdAT2 organoids could differentiate in a more physiological environment. We dissected adult mouse lungs and injected them

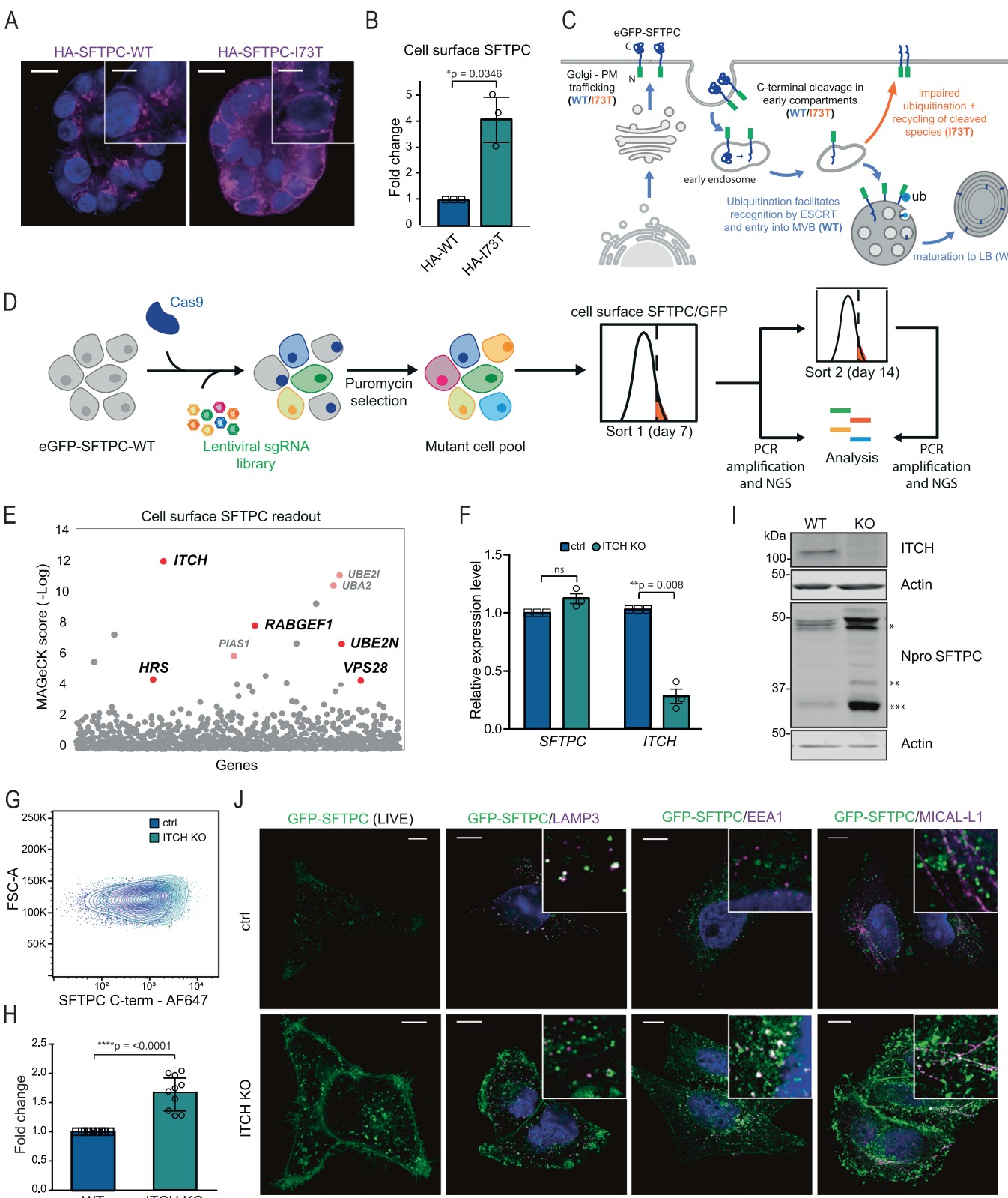

◄

**Figure 3.   A forward genetic screen confirms the importance of ubiquitination in SFTPC trafficking and identifies the E3 ligase ITCH as required for SFTPC maturation.**

(A) Immunofluorescence of proSFTPC localization in fdAT2 organoids expressing HA-SFTPC-WT or HA-SFTPC-I73T for 10 days. Scale bar, 10 μm/5 μm in zoomed inserts. (B) Quantification of cell surface SFTPC in organoids expressing HA-SFTPC-WT or HA-SFTPC-I73T as measured by SFTPC C-terminal antibody, expressed as fold change in mean fluorescence intensity; mean ± SD, $n = 3$ biologically independent organoid lines (one-sample $t$ test). (C) Schematic of WT and pathogenic I73T variant SFTPC trafficking. (D) Schematic of ubiquitome forward genetic screen strategy. (E) MAGeCK score demonstrating relative enrichment of each gRNA for the day 7 cell surface SFTPC-high population. (F) Relative expression of *SFTPC* and *ITCH* in control and *ITCH* knockout (KO) HeLa cells as measured by RT-qPCR and normalized to GAPDH; mean ± SD, $n = 3$ independent repeats (one-sample $t$ test). (G, H) Flow cytometry of cell surface SFTPC as measured by SFTPC C-terminal (C-term) antibody in control vs *ITCH* knockout lines and quantification of mean fluorescence intensity; mean ± SD, $n = 10$ independent repeats (one-sample $t$ test); ****$P = 0.0000472$. (I) Immunoblotting of lysates from control or *ITCH* knockout cells using an SFTPC N-terminal (Npro) and ITCH antibody ( = full-length species ( ± palmitoylation), **partially C-terminal cleaved intermediate, ***fully C-terminally cleaved intermediate). (J) Live cell imaging (left panel) and colocalisation of GFP-SFTPC with LAMP3, MICALL1 and EEA1 by immunofluorescence in control and *ITCH* knockout cells. Scale bar, 10 μm/5 μm in zoomed inserts. See also Appendix Figs. S2–3. Source data are available online for this figure.

with single cells isolated from lentiviral-transduced *SFTPC-GFP;EF1a*-RFP human fdAT2 organoids, followed by precision-cut lung slice (PCLS) culture for 1 week (Fig. 2H). The RFP⁺ human cells were engrafted into the alveolar structure in the mouse PCLS and showed flattened nuclei, consistent with reports that AT1 cells have flattened nuclear shape (Shiraishi et al, 2023) (Fig. 2I,J; Movies EV2–4). Scoring for the *SFTPC*-GFP reporter and AT1/2 fate markers confirmed that nearly 100% of the RFP⁺ cells co-expressed the AT1 lineage markers, CAV1 and AGER, and rarely expressed the *SFTPC*-GFP reporter or SFTPC protein (Fig. 2I–K). Taken together, these data show that the human fdAT2 organoids are competent to differentiate to the AT1 cell lineage in vitro and in the mouse lung environment ex vivo.

Our data indicate that the fdAT2 organoids self-renew, differentiate to AT1 cells and display a mature surfactant synthesis profile during prolonged passaging. They are amenable to lentiviral transduction and therefore represent a physiological system to study mechanisms of fundamental AT2 function and dysfunction in disease.

The commonest pathogenic variant of SFTPC, I73T, mislocalises to the plasma membrane and causes AT2 dysfunction via a toxic gain-of-function effect leading to ILD (Brasch et al, 2004; Alysandratos et al, 2021). We were able to reproduce this cell surface phenotype in the fdAT2 organoids following viral transduction of HA-SFTPC variants (Fig. 3A,B). Ubiquitinated SFTPC is recognized by the ESCRT machinery and trafficked to late compartments. We previously showed that the disease-causing SFTPC-I73T mutant is no longer ubiquitinated, resulting in its relocalisation via recycling and saturated endocytosis of immature isoforms (Dickens et al, 2022) (Fig. 3C). To identify the ubiquitination machinery required for SFTPC maturation, we performed a targeted forward genetic screen (Figs. 3D and EV4A). We predicted that depletion of key effectors of SFTPC ubiquitination would phenocopy the I73T variant by causing SFTPC accumulation at the plasma membrane. HeLa cells stably expressing GFP-SFTPC and Cas9 were transduced with a subgenomic ubiquitome sgRNA library (Menzies et al, 2018), and cells with increased surface-localized proSFTPC compared with untransduced controls were harvested at day 7 or 14 (Fig. EV4B,C). Transduced cells were also sorted for total SFTPC (GFP-high) to ensure that any cells accumulating C-terminally cleaved SFTPC at the cell surface (and thus missing the antibody epitope, Fig. EV4A) were not overlooked (Fig. EV4D,E).

The most enriched gRNA in d7 cell surface SFTPC-high cells targeted Itchy E3 Ubiquitin Protein Ligase (*ITCH*) (Figs. 3E

and EV4B; Dataset EV4). Guide RNAs targeting ESCRT machinery components (hepatocyte growth factor-regulated tyrosine kinase substrate (*HRS*) and VPS28 subunit of ESCRT-I (*VPS28*)), K63-chain-specific Ubiquitin E2-Conjugating Enzyme E2 N (*UBE2N*) and the early endosome Rab5-specific GEF RAB Guanine Nucleotide Exchange Factor 1 (*RABGEF1*) were also enriched; supporting our previous data that SFTPC transits early endosomal compartments before K63-ubiquitination, recognition by the ESCRT complex and transit into MVBs. These hits largely overlapped with the GFP-defined sort which also included gRNAs targeting *ITCH* and ESCRT machinery (Fig. EV4D). Enrichment of gRNAs for SUMOylation-related Ubiquitin Conjugating Enzyme E2 I (*UBE2I*), Ubiquitin Like Modifier Activating Enzyme 2 (*UBA2*) and Protein Inhibitor Of Activated STAT 1 (*PIAS1*) was noted in both screens. *ITCH* remained highly enriched in the d14 sort, though other specific hits (e.g., *HRS*, *VPS28*, *UBE2N*) were lost likely due to their fundamental roles in trafficking and thus cellular toxicity when depleted (Fig. EV4C,E).

Initial validation of *ITCH*, SUMOylation components and positive controls *HRS* and *UBE2N* (Appendix Fig. S2A) revealed that individual depletion resulted in marked cell surface GFP-SFTPC localization (Appendix Fig. S2B). Enrichment of surface-localized full-length SFTPC was more modest (Appendix Fig. S2C), suggesting that a proportion of surface-localized SFTPC has lost the epitope for the antibody used for flow cytometry through C-terminal cleavage. We used RT-qPCR to test the hypothesis that SUMOylation hits reflected changes in *SFTPC* transcription (Rosonina et al, 2017), rather than trafficking (Fig. EV5A). Further investigation using the SUMO-activating enzyme inhibitor TAK-981 revealed that manipulation of SUMOylation affected SFTPC expression and localization only when under the control of a pCMV promoter and had no effect on endogenous SFTPC in fdAT2 organoids (Fig. EV5B–K).

ITCH is a HECT-type E3 ligase whose WW domains recognize cytosolic proline-rich consensus sequences, typically PPxY which is present within the N-terminal tail of SFTPC (Chen and Sudol, 1995). Depletion of *ITCH* in clonal GFP-SFTPC expressing HeLa lines did not affect *SFTPC* transcription (Fig. 3F), but markedly increased plasma membrane resident SFTPC (Fig. 3G–J). Immunoblotting revealed an excess of full-length (*), partially cleaved (**) and fully C-terminally cleaved (***) species, suggesting that ITCH depletion inhibits trafficking at/beyond the location of the final C-terminal cleavage which is thought to be at the MVB limiting membrane (Dickens et al, 2022; Beers et al, 1994) (Fig. 3I). We confirmed that SFTPC resides largely in LAMP3⁺ (late)

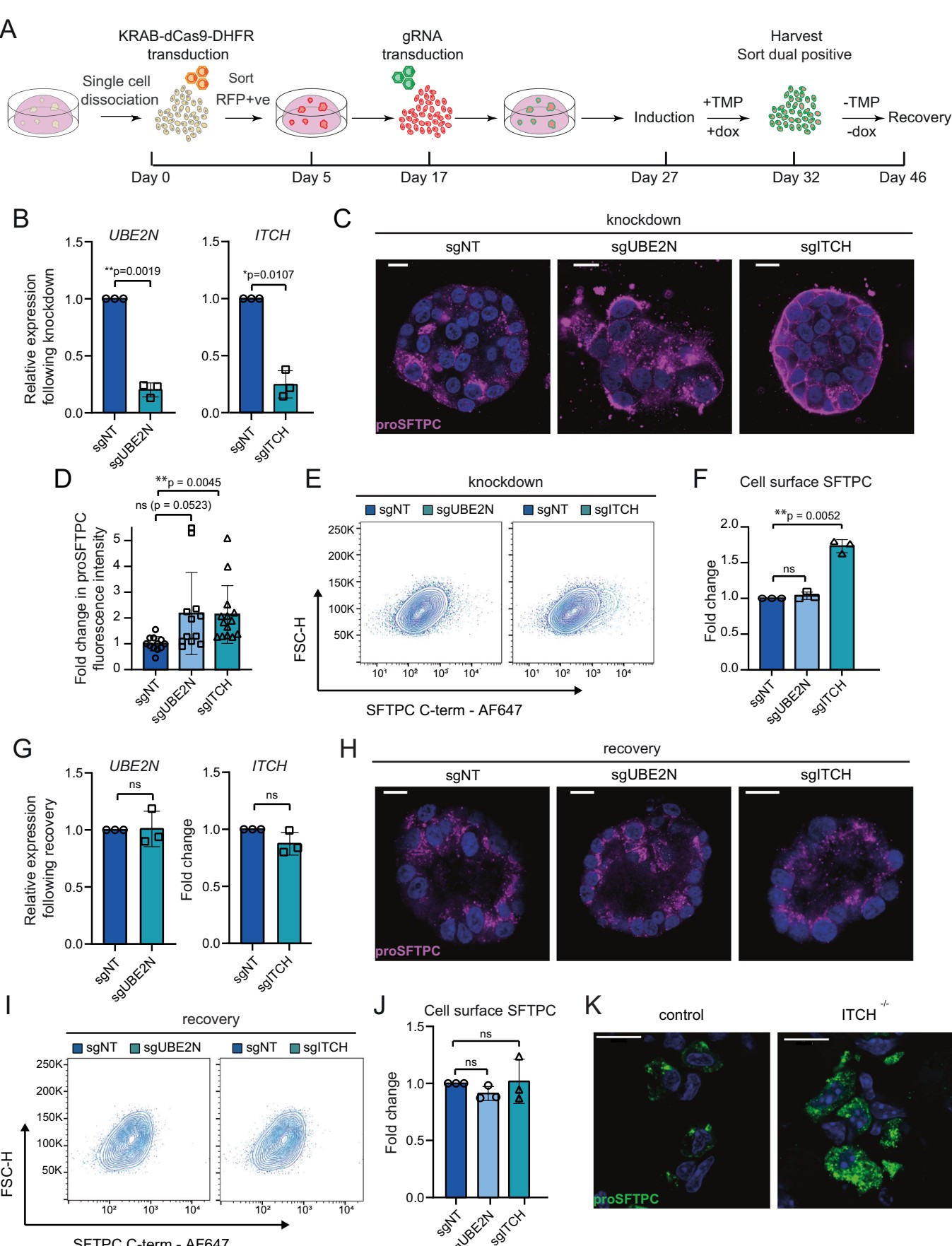

**Figure 4.    ITCH depletion alters SFTPC localization in fetal lung-derived AT2 organoids.**

(A) Schematic of the inducible CRISPR interference (CRISPRi) system in fdAT2 organoids. (B) Relative expression of *UBE2N* and *ITCH* following CRISPRi induction. Mean ± SD, $n = 3$ biologically independent organoid lines (one-way ANOVA with Tukey multiple comparison post-test). (C) ProSFTPC localization in CRISPRi-depleted organoids as measured by immunofluorescence with exposure optimized to show SFTPC subcellular localization. Scale bar, 10 μm. (D) Quantification of proSFTPC abundance in CRISPRi-depleted organoids as measured by fluorescence intensity. Mean ± SD; minimum 3 organoids per line and minimum 3 biologically independent lines (Brown-Forsythe and Welch ANOVA with Dunnetts comparison). (E, F) Quantification of cell surface full-length SFTPC as measured by flow cytometry using SFTPC C-terminal antibody, shown as representative contour plot (E) and expressed as fold change in mean fluorescence intensity; mean ± SD, $n = 3$ biologically independent organoid lines (one-sample *t* test) (F). (G–J) RT-qPCR, microscopy and flow cytometry analysis of organoids following a 14-day dox and TMP withdrawal recovery period; mean ± SD, $n = 3$ biologically independent organoid lines (one-sample *t* test). Scale bar, 10 μm (zoom 5 μm). (K) Intracellular localization of SFTPC in wild-type and *Itch* knockout mice ($n = 2$). Scale bar, 10 μm. Source data are available online for this figure.

compartments in control cells. Following ITCH depletion, SFTPC relocalised to EEA1$^+$ early endosomes and MICALL1$^+$ recycling endosomes, consistent with failure of MVB entry and recycling to the plasma membrane (Fig. 3J). In rescue experiments, restoration of ITCH in knockout cells reversed SFTPC mislocalisation (Appendix Fig. S3). These data suggest that ITCH is required for SFTPC trafficking and that ITCH depletion causes relocalisation of SFTPC to the plasma membrane, phenocopying SFTPC-I73T.

Having identified ITCH as an E3 ligase required for SFTPC maturation using HeLa cells, we wanted to determine whether ITCH also regulates endogenous SFTPC trafficking in primary AT2 cells. We therefore confirmed that these transcripts were expressed in fdAT2 organoids (Fig. EV3K) and then used CRISPRi to genetically deplete fdAT2 organoids of ITCH and positive control UBE2N. CRISPRi-expressing fdAT2 organoids were transduced with a gRNA for each gene and silencing induced (Sun et al, 2021) (Fig. 4A). Approximately 50–80% of organoids expressed both the CRISPRi and gRNA after 5 days of induction (Appendix Fig. S4B). The double-transduced population typically achieved 80–90% gene silencing (Fig. 4B). Transcription of *SFTPC* was unaffected by *ITCH* or *UBE2N* depletion (Appendix Fig. S4A). We also investigated HECT domain E3 ligase, NEDD4-2, which has been reported to play a role in SFTPC ubiquitination and maturation (Kotorashvili et al, 2009; Duerr et al, 2020) but was not isolated in our screen. However, *NEDD4-2* was resistant to depletion and not pursued further (Appendix Fig. S4C).

Targeted inhibition of ubiquitination via *UBE2N* or *ITCH* depletion resulted in accumulation of intracellular SFTPC with marked relocalization to the plasma membrane (Fig. 4C,D; Appendix Fig. S4D). Interestingly, this was detectable as excess full-length SFTPC by flow cytometry following *ITCH* depletion (Fig. 4E,F). *UBE2N* depletion did not have this effect suggesting the accumulated protein is C-terminally cleaved, implicating its role beyond early endosomes where initial proteolysis occurs, whereas ITCH likely mediates its effect earlier in the trafficking pathway before initial SFTPC cleavage. However, the excess of full-length protein may also reflect a profound effect of *ITCH* depletion on onward SFTPC trafficking and saturation of endocytic compartments. The knock-down effects reversed completely if organoids were recovered without silencing (-TMP/-dox) (Fig. 4G–J). Depletion of SFTPC ubiquitin ligases in human fdAT2 organoids thus results in mislocalization of endogenous SFTPC.

ITCH deficiency causes lung interstitial inflammatory infiltrates in humans and mouse models (Lohr et al, 2010; Aki et al, 2018) thought to result from multisystem autoimmune disease. Though fdAT2 are unlikely to reproduce immune perturbations, we predicted an additional non-immune mediated alveolar epithelial

phenotype if ITCH is important for SFTPC maturation in vivo. Staining for proSFTPC revealed intracellular accumulation and an altered distribution in *Itch* deficient (*Itch*$^{a18H/a18H}$) mouse alveolar epithelium. Numerous, smaller proSFTPC puncta are consistent with failure of trafficking to late compartments, seen as large puncta in the wild-type mice (Fig. 4K).

We conclude that intracellular trafficking of SFTPC requires K63 ubiquitination mediated by the HECT domain E3 ligases including ITCH. Inhibiting ubiquitination phenocopies the redistribution of the pathogenic SFTPC-I73T variant, reinforcing the importance of this post-translational modification in maintaining AT2 health.

## Discussion

We have derived an expandable AT2 organoid model from fetal lung tip progenitor cells and demonstrated its use in investigating mechanisms of AT2 biology relevant to disease. FdAT2 organoids acquire features of mature adult AT2 cells, including lamellar body formation, mature surfactant protein secretion, and the ability to differentiate to AT1 cells (Figs. 1 and 2). The fdAT2 organoids readily expand, can be passaged multiple times and cryopreserved without compromising their identity. They are also highly amenable to genetic manipulation, enabling us to investigate key SFTPC trafficking effectors using CRISPRi (Figs. 3 and 4).

Complete trafficking and maturation of SFTPC is required for AT2 health; misfolding variants are retained in early compartments and cause ER stress Mulugeta et al, 2005; Katzen et al, 2019, whereas mistrafficking isoforms, exemplified by SFTPC-I73T, mislocalise due to a trafficking block and failure of ubiquitination (Alysandratos et al, 2021; Dickens et al, 2022). Both mutation types result in AT2 cell dysfunction and heritable forms of interstitial lung disease. These phenotypes have been confirmed in animal models (Katzen et al, 2019; Nureki et al, 2018; Bridges et al, 2003; Lawson et al, 2011), but mechanistic work on AT2 dysfunction in physiological human ex vivo models has been highly challenging. We combined a forward genetic screen with genetic manipulation of fdAT2 organoids to investigate intracellular SFTPC trafficking via the plasma membrane. We confirm that ubiquitination is required for normal SFTPC trafficking and identify the E3 ligase ITCH as a novel effector of SFTPC processing. We further confirmed altered SFTPC handling in *Itch* deficient mice (Fig. 4K).

FdAT2 organoids are an attractive genetic model for understanding fundamental mechanisms of AT2 cell biology in health, and modeling inherited disorders and environmental insults which perturb their function. They also represent a useful cell source to investigate cellular and molecular mechanisms of AT2 to AT1 cell

lineage differentiation, and AT2 plasticity, during human lung development and repair.

# Methods

### Reagents and tools table

| Reagent/resource | Reference or source | Identifier or catalog number |
|---|---|---|
| **Experimental models (species)** | | |
| HeLa (*H. sapiens*) | ATCC | CCL-2 |
| HEK-293T (*H. sapiens*) | Gift from Paul Lehner, University of Cambridge | N/A |
| B6 Itch^a18H/a18H (*M. musculus*) | Dr. Lydia Matesic, University of South Carolina | N/A |
| C57BL/6J (*M. musculus*) | Gurdon Institute, University of Cambridge | N/A |
| **fdAT2 organoid lines (*H. sapiens*)** | | |
| HDBR-L 16393 | This study | N/A |
| HDBR-L 15934 | This study | N/A |
| HDBR-L 16392 | This study | N/A |
| HDBR-L 17873 | This study | N/A |
| HDBR-L 16402 | This study | N/A |
| HDBR-L 16587 | This study | N/A |
| HDBR-L 16011 | This study | N/A |
| HDBR-L 17884 | This study | N/A |
| **Recombinant DNA** | | |
| **Forward genetic screen** | | |
| pCMVR8.91 (Gag-Pol) | Gift from Paul Lehner, University of Cambridge | N/A |
| pMD2G (VSV-G) | Addgene | 12259 |
| pHRSIN.pSFFV 3FLAG-CAS9 pGK Hygro | Gift from Paul Lehner, University of Cambridge | Timms et al, 2016 |
| pKLV-pU6-esgRNA(modified BbsI)-pPGK-Puro2ABFP | Addgene | 50946 |
| **Organoid transduction** | | |
| psPAX2 (Gag-Pol) | Addgene | 12260 |
| pAdvantage | Promega | E1711 |
| pLenti-tetON-KRAB-dCas9-DHFR-EF1a-TagRFP-2A-tet3G | Addgene | 1679351 |
| pLenti-U6-gRNA-EF1a-EGFP-CAAX | Addgene | 1679361 |
| pLenti-hSFTPCpromoter(2 kb)-EGFP-EF1a-TagRFP | Addgene | 201681 |
| pLenti-hSCGB3A2-EGFP-EF1a-TagRFP | Addgene | 204821 |

| Reagent/resource | Reference or source | Identifier or catalog number |
|---|---|---|
| **CRISPR-Cas9 mediated gene knockout** | | |
| pSpCas9(BB)-2A-mCherry | Gift from David Ron, University of Cambridge | N/A |
| **HeLa rescue** | | |
| pCR3.1-ITCH | Gift from Juan Carlos, Universidad de Salamanca | N/A |
| **SFTPC-WT/I73T expression** | | |
| pLenti-tetON-HA-SFTPC-EF1a-tagRFP-2A-tet3G | This paper; deposited at Addgene | 201681 |

| Antibodies | Concentration | Reference or source | Identifier or catalog number |
|---|---|---|---|
| **Immunofluorescence** | | | |
| Rabbit anti-proSFTPC N terminus | 1:200 | Abcam | Ab90716 |
| Mouse anti-EEA1 | 1:200 | BD biosciences | BD610457 |
| Mouse anti-Mical-L1 | 1:100 | Novus | H00085377 |
| Mouse anti-LAMP3 | 1:200 | DHSB | H5C6 |
| Mouse anti-HA-488 conjugate | 1:200 | Invitrogen | A-21287 |
| Rabbit anti-proSFTPC | 1:200 | Merck | AB3786 |
| Rat anti-E-cadherin | 1:500 | Thermo Fisher Scientific | 13-1900 |
| Rabbit anti-LAMP3 | 1:100 | Atlas Antibodies | HPA051467 |
| Mouse anti-HTII-280 | 1:200 | Terrace Biotech | TB-27AHT2-280 |
| Rabbit anti-mature SFTPC | 1:200 | Seven Hills Bioreagents | WRAB-76694 |
| Rabbit anti-mature SFTPB | 1:200 | Seven Hills Bioreagents | WRAB-48604 |
| Mouse anti-ABCA3 | 1:200 | Seven Hills Bioreagents | WMAB-17G524 |
| Mouse anti-NAPSA | 1:200 | Novus Biological | NBP2-45245 |
| Rabbit anti-ZO-1 | 1:200 | Thermo Fisher Scientific | 40-2200 |
| Mouse anti-β-catenin | 1:200 | BD science | 610154 |
| Rabbit anti-pan-Laminin | 1:400 | Abcam | Ab11575 |
| Rabbit anti-CAV1 | 1:200 | Cell Signaling Technologies | 3267S |
| Rabbit anti-AGER | 1:200 | Proteintech | 16346-1-AP |
| Mouse anti-β-actin | 1:200 | Merck | A3854 |
| Mouse anti-HOPX | 1:200 | Santa Cruz | Sc-398703 |
| Goat anti-SOX9 | 1:200 | R&D systems | AF3075 |
| Rabbit anti-ZO-1 | 1:100 | Abcam | ab221547 |
| Rabbit anti-HOPX | 1:100 | Abcam | ab307670 |

| Reagent/resource | Reference or source | | Identifier or catalog number |
| --- | --- | --- | --- |
| Mouse anti-Integrin beta 1 | 1:200 | Abcam | ab30394 |
| AlexaFluor Plus 647 Phalloidin | 1:400 | Invitrogen | 16400815 |
| AlexaFluor™ 488 phalloidin | 2 drops/mL | Invitrogen | 14879680 |
| Donkey anti-Mouse IgG (H + L) AlexaFluor™ 488 | 1:1000 | Invitrogen | 10544773 |
| Donkey anti-Goat IgG (H + L) AlexaFluor™ 488 | 1:1000 | Invitrogen | 10246392 |
| Donkey anti-Rabbit IgG (H + L) AlexaFluor™ 647 | 1:1000 | Invitrogen | 10543623 |
| Donkey anti-Rat IgG (H + L) AlexaFluor™ 594 | 1:1000 | Invitrogen | 10133952 |
| **Immunoblotting** | | | |
| Rabbit anti-proSFTPC N terminus | 1:1000 | Abcam | Ab90716 |
| Mouse anti-ITCH | 1:1000 | Proteintech | 67757-1-IG |
| Rabbit anti-mature SFTPC | 1:1000 | Seven Hills Bioreagents | WRAB-76694 |
| Rabbit anti-mature SFTPB | 1:1000 | Seven Hills Bioreagents | WRAB-48604 |
| Mouse anti-GAPDH | 1:2000 | Abcam | Ab8245 |
| Mouse anti-β-actin | 1:10,000 | Abcam | Ab3280 |
| *Anti-mouse IRDye® 800CW* | 1:20,000 | Abcam | Ab216774 |
| *Anti-rabbit IRDye®800CW* | 1:20,000 | Abcam | Ab216773 |
| *Anti-rabbit IRDye® 680RD* | 1:20,000 | Abcam | Ab216779 |
| *Anti-mouse IRDye® 680RD* | 1:20,000 | Abcam | Ab216776 |
| **Flow cytometry** | | | |
| Rabbit anti-proSFTPC C terminus | 1:100 | Abgent | AP-13684b |
| Mouse anti-pan-HLA W6/32 | 1:10 | Made in-house by Lehner lab, University of Cambridge | N/A |
| Rabbit anti-EpCAM | 1:25 | Abcam | Ab223582 |
| Mouse anti-HTII-280 | 1:25 | Terrance Biotech | TB-27AHT2-280 |

| Oligonucleotides and other sequence-based reagents | Reference or source | Identifier or catalog number |
| --- | --- | --- |
| qRT-PCR primers | This study | Appendix Table 1 |
| gRNA sequences | This study | Appendix Table 2 |

| Reagent/resource | Reference or source | Identifier or catalog number |
| --- | --- | --- |
| **Chemicals, enzymes, and other reagents** | | |
| TAK-981 | Caymen Chemicals | 32741 |
| Collagenase | Merck | C9891 |
| Dispase | Thermo Fisher Scientific | 17105041 |
| DNase | Merck | D4527 |
| RBC lysis buffer | BioLegend | 420301 |
| EpCAM (CD326) microbeads | Miltenyi Biotec | 130-061-101 |
| Matrigel | Corning | 356231 |
| Y-27632 | Merck | 688000 |
| A83-01 | Tocris | 2939 |
| CHIR99021 | Stem Cell Institute, University of Cambridge | N/A |
| Recombinant human FGF7 | PeproTech | 100-19 |
| Recombinant human FGF10 | PeproTech | 100-26 |
| DAPT | Merck | D5942 |
| 3-Isobutyl-1-methylxanthine (IBMX) | Merck | 15679 |
| 8-Bromoadenosine 3'5'-cyclic monophosphate (cAMP) | Merck | B5386 |
| Dexamethasone | Merck | D4902 |
| N-acetylcysteine | Sigma | A9165 |
| N2 supplement | Gibco | 17502048 |
| B27 supplement | Gibco | 17504-44 |
| Penicillin/Streptomycin | Gibco | 15630-080 |
| HEPES | Gibco | 15140-122 |
| GlutaMax | Gibco | 35050-061 |
| Advanced DMEM/F12 | ThermoFisher | 12634010 |
| DMEM | Sigma-Aldrich | D6429 |
| Fetal bovine serum (FBS) | Sigma-Aldrich | F7524 |
| Human serum | Merck | H4522 |
| FuGene 6 | Promga | E2691 |
| Lipofectamine LTX | ThermoFisher | A12621 |
| Hygromycin | Invitrogen | 10687010 |
| Puromycin | Alfa Aesar | J61278 |
| Trimethoprim (TMP) | Merck | 92131 |
| Doxycycline | Merck | D9891 |
| LATS-IN-1 | Cambridge Bioscience | CAY36623 |
| Lenti-X | Takara bio | 631231 |
| DAPI | Merck | D9542 |

| Reagent/resource | Reference or source | Identifier or catalog number |
|---|---|---|
| Prolong Gold antifade | Invitrogen | P36934 |
| HistoGel | Epredia | HG-4000-012 |
| Cell recovery solution | Corning | 354253 |
| 5% normal donkey serum | Stratech Scientific | 017-000-121-JIR |
| 2′ − 2′-thio-diethanol (TDE) | Merck | 166782 |
| Lysotracker | Cell Signaling Technologies | 8783S |
| Poly-D-Lysine | Biotechne | 3439-100-01 |
| PBS-Tween-20 | Promega | H5151 |
| VECTASHIELD® PLUS Antifade Mounting Medium | Vector laboratories | H-1900 |
| SYBR Green JumpStart Taq Readymix | Sigma-Aldrich | S4438 |
| SYBR Green PCR Master Mix | Applied Biosystems | 4309155 |
| TrypLE Express enzyme | Thermo Fisher Scientific | 12605010 |
| **Kits** | | |
| High-capacity cDNA reverse transcription kit | Applied Biosystems | 4368814 |
| RNeasy kit | Qiagen | 74004 |
| Puregene Core Kit | Qiagen | 1042601 |
| Chromium Single Cell 5′ Reagents Kit | 10x Genomics | N/A |
| **Software** | | |
| GraphPad Prism (version 9.5.1) | RRID:SCR_002798 | |
| MAGeCK | RRID:SCR_025016 Li et al, 2014 | |
| ZEISS ZEN Microscopy Software | RRID:SCR_013672 | |
| FlowJo software (10.0.0) | RRID:SCR_008520 | |
| STARsolo 2.7.10a | Kaminow et al, 2021 | |
| EmptyDrops | Lun et al, 2019 | |
| cellsnp-lite | Huang et al, 2021 | |
| souporcell | Heaton et al, 2020 | |
| Seurat package (v4.0) | | |
| **Other** | | |
| Glass bottom microwell dishes | MatTek | P35G-1.5-14-C |
| 0.4 µm transwell inserts | Greiner Bio-One | 662641 |
| AMPure XP magnetic beads | Beckman Coulter | A63881 |

| Reagent/resource | Reference or source | Identifier or catalog number |
|---|---|---|
| Illumina MiniSeq | Illumina | |
| Illumina NovaSeq 6000 | Illumina | |
| SP8 confocal microscope | Leica | |
| Zeiss LSM880 with airyscan | Zeiss | |
| Odyssey Imager | Li-Cor | |
| FACSMelody | BD bioscience | |
| SH800S Cell sorter | Sony Biotechnology | |
| LSRFortessa | BD bioscience | |
| Tecnai transmission electron microscope | FEI | |
| Agilent 2200 Tapestation | Agilent | |

## Methods and protocols

### Material availability

Human organoid lines used in the study are available from Dr Emma L Rawlins (elr21@cam.ac.uk) with a completed Materials Transfer Agreement.

### Mouse tissue

In all, 6–10 week-old male C57BL/6 mice were used for the ex vivo AT1 differentiation experiments. All procedures were approved by the University of Cambridge Animal Welfare and Ethical Review Body and carried out under a UK Home Office License (PPL: PP3176550) in accordance with the Animals (Scientific Procedures) Act 1986. Mice were bred and maintained under specific-pathogen-free conditions at the Gurdon Institute of the University of Cambridge.

Twelve-week-old female animals homozygous for a null allele of *Itch* (*Itch*$^{a18H/a18H}$, B6.C3H(101)-In(2a;Itch)18H/LmatMmjax MMRRC stock #65285) have been previously described (Hustad et al, 1995; Perry et al, 1998). For this line, the $a^{18H}$ allele was backcrossed to C57BL/6J for 27 generations. Therefore, age- and gender-matched C57BL/6J mice were used as controls (*Itch*$^{+/+}$) in the indicated experiments. No blinding was performed. All mice were cared for in accordance with the National Institute of Health's Guide for the Care and Use of Laboratory Animals (8th edn, Washington, DC: 2011) and the University of South Carolina's Institutional Animal Care and Use Committee approved all experimental protocols. Mice were bred and maintained under specific-pathogen-free conditions at the University of South Carolina.

### Human embryonic and fetal lung tissues

Human embryonic and fetal lung tissue was provided from terminations of pregnancy from Cambridge University Hospitals NHS Foundation Trust under permission from NHS Research Ethical Committee (96/085) and the MRC/Wellcome Trust Human

Developmental Biology Resource (London and Newcastle, University College London (UCL) site REC reference: 18/LO/0822; Project 200591; www.hdbr.org). Sample age ranged from 7 to 9 and from 16 to 22 weeks of gestation (postconception weeks; pcw). Sample gestation was determined by external physical appearance and measurements. Samples had no known genetic abnormalities. The sample gender was unknown at the time of collection and was not determined. All collected samples were included in the study.

### Derivation and in vitro culture of human alveolar type 2-like organoids

Distal edges of 16–22 pcw human fetal lung tissue, typically measuring 0.5 cm × 1–3 cm, were isolated and fragmented into smaller pieces (Nikolić et al, 2017). Fragments were dissociated with 0.125 mg/ml Collagenase, 1 U/ml Dispase, and 0.1 U/μl DNase in a rotating incubator for 1 h at 37 °C. After rinsing in washing buffer containing 2% FBS in cold PBS the cells were filtered through a 100-μm strainer. Cells were treated with RBC lysis buffer at room temperature for 5 min, rinsed and incubated with EpCAM (CD326) microbeads to isolate EpCAM+ epithelial cells containing mostly tip epithelial cells. These cells were embedded in Matrigel and cultured in 24-well plates in alveolar type 2 differentiation medium (AT2 medium): Advanced DMEM/F12 supplemented with 1× GlutaMax, 1 mM HEPES and penicillin/streptomycin, 1× B27 supplement (without Vitamin A), 1× N2 supplement, 1.25 mM N-acetylcysteine, 50 nM Dexamethasone, 0.1 mM 8-Bromoadenosine 3'5'-cyclic monophosphate, 0.1 mM 3-Isobutyl-1-methylxanthine (IBMX), 50 μM DAPT, 100 ng/ml recombinant human FGF7, 3 μM CHIR99021, 10 μM A83-01, and 10 μM Y-27632. Medium was replaced every 2 days and cultures maintained for 2 weeks until the initial fdAT2 organoid colonies formed in the matrigel droplet. Organoids were typically passaged at a 1 to 3 ratio weekly by gently breaking into small fragments. Organoid lines were split equally between different experimental conditions at passaging, and no blinding was performed. All organoid experiments were performed with organoids obtained from at least three different biological donors, except in Fig. EV1B where 1 biological donor was used. No sample size estimates were performed.

For *SFTPC*-GFP (this manuscript), or *SCGB3A2*-GFP (He et al, 2022) reporter cell generation, the isolated tip progenitor cells or proximal airway cells were immediately transduced and selected based on *SFTPC*-GFP or *SCGB3A2*-GFP with *EF1a*-TagRFP reporter expression by flow cytometry, after 48 h of transduction and cultured for 3 weeks in the AT2 medium. As a control, proximal airway progenitors were immediately selected based on *SCGB3A2*-GFP after 48 h of transduction and cultured in airway medium: Advanced DMEM/F12 medium supplemented with 1× B27 (without Vitamin A), 1× N2, 1.25 mM N-acetylcysteine, 100 ng/ml FGF10, 100 ng/ml FGF7, 50 nM Dexamethasone, 0.1 mM cAMP, 0.1 mM IBMX, and 10 μM Y-27632.

### Immortalized cell culture and cell line derivation

HeLa cells were cultured in DMEM + 10% fetal bovine serum (FBS). Plasmid DNA was introduced using liposomal transfection with FuGene 6 or lipofectamine LTX.

Clonal GFP-SFTPC-expressing HeLa cell lines were derived as previously described (Dickens et al, 2022). A pool of HeLa cells stably expressing Cas9 was generated by lentiviral transduction of the pHRSIN-Cas9 vector and selection of transduced cells using hygromycin (2 μg/ml). For inhibition of the SUMOylation pathway 25–500 nM TAK-981 (Langston et al, 2021) was added to the medium.

For initial CRISPR screen validation, gene-specific deplete pools were obtained by transducing HeLa cells with pKLV-pU6-esgRNA(modified BbsI)-pPGK-Puro2ABFP containing 3 gRNAs per gene and transduced cells selected with puromycin (1 μg/ml). *ITCH*-depleted clonal lines were subsequently derived by single-cell sorting into 96-well plates before expansion and screening by RT-qPCR. Experiments were performed on a minimum of three clonal lines derived from different gRNAs to ensure findings did not reflect off-target effects.

For ITCH rescue experiments, four silent mutations were introduced into the *ITCH* cDNA sequence in the region of the gRNA by site-directed mutagenesis (GA**A** CGG CGG GT**T** GA**C** AA**C** ATG - > GA**G** CGG CGG GT**A** **G**A**T** AA**T** ATG) before transfection to ensure the integrated CRISPR-Cas9 in the ITCH knockout cells did not also edit the transiently transfected plasmid.

To determine the efficacy of candidate CRISPRi gRNAs, HeLa cells were transduced with pLenti-tetON-KRAB-dCas9-DHFR-EF1a-TagRFP-2A-tet3G and sorted by FACS. Expression was induced with 10 nmol/l trimethoprimand 2 μg/ml doxycycline for 5 days before cells were harvested for RNA and RT-qPCR.

### Alveolar type 1 (AT1) lineage differentiation of fdAT2 organoids in vitro and ex vivo

For in vitro AT1 differentiation, fdAT2 organoids were either (1) dissociated into single cells and $1 \times 10^5$ cells were replated on matrigel-coated 12-well dishes and cultured in an AT1-promoting medium containing 10% human serum for 2 weeks (Katsura et al, 2020), or (2) re-embedded in matrigel and cultured in 50 nM Dexamethasone, 0.1 mM cAMP, 0.1 mM IBMX, 100 ng/ml recombinant human FGF7, 10 μM LATS1 and 2 inhibitor LATS-IN-1 in the absence of CHIR99021 for a week (Burgess et al, 2024). Control medium substituted DMSO for LATS-IN-1. Both media are based on Advanced DMEM/F12 medium supplemented with 1× B27 (without Vitamin A), 1× N2, 1.25 mM N-acetylcysteine, 1× penicillin/streptomycin. Cells were analyzed by RT-qPCR and immunofluorescence.

AT1 lineage differentiation of fdAT2 organoids was also performed in an ex vivo mouse lung explant. Prior to the ex vivo culture of precision-cut lung slices (PCLS) (Rosales Gerpe et al, 2018), fdAT2 organoids expressing *SFTPC*-GFP+ and TagRFP+ were dissociated into single cells, $2 \times 10^5$ cells mixed with 100 μl 50% matrigel and directly injected into each lung lobe using an 18 G needle. Mouse lung lobes were sectioned into 150 μm PCLS using a Leica VT1200s vibratome. Following ex vivo culture in DMEM/F12 medium supplemented with 2% v/v penicillin–streptomycin, 10% FBS and 1 mM cAMP for 1 week, the PCLS were subjected to immunofluorescence analysis.

### Lentivirus production and viral transduction

Lentivirus was produced using HEK-293T cells. For pHRSIN and pKLV vectors (for the forward genetic screen), $5 \times 10^5$ cells were plated in one well of a six-well plate 24 h before transfection with 0.67 μg pCMVR8.91 (Gag-Pol), 0.33 μg pMD2G (VSV-G) and 1 μg pHRSIN/pKLV (with gene of interest) and media was changed the next day. Virus was collected after 48 h and filtered through a 0.45-μm filter. For

pLenti vectors for the CRISPRi experiments and generating *SFTPC*-GFP reporter line, $2 \times 10^5/8 \times 10^5$ cells ($< 10$ kb / $> 10$ kb plasmid, respectively) were plated into a 10-cm dish 24 h before transfection with 5/10 μg plasmid of interest, 3/6 μg pPAX2, 2/3 μg pMD2.G and 2/3 μg pAdvantage and media was changed the next day. Virus was collected after 48 h and filtered through a 0.45-μm filter before typically being concentrated with Lenti-X concentrator (3:1 supernatant to Lenti-X) overnight at 4 °C, spun at $1500 \times g$ for 45 min in a cold centrifuge and the pellet resuspended such that virus was ×100 concentrated.

Transduction of HeLa cells was typically achieved by adding 500 μl unconcentrated viral supernatant to $2 \times 10^5$ cells in a six-well plate, centrifuging at $600 \times g$ for 1 h and incubating overnight before refreshing the media. Antibiotic selection was added 48 h after transduction.

### Genetic manipulation of organoids

To create transduced organoid lines, single-cell dissociated fdAT2 organoids were typically infected with 25 μl 100× concentrated lentivirus overnight at 37 °C then embedded into matrigel and cultured in the medium for another 48–72 h before fluorescence sorting to enrich for transduced cells. For the reporter system, pHAGE plasmids were modified by insertion of EGFP or EF1a-promoter TagRFP (EF1a-TagRFP) cassettes into a shortened fragment of the human *SFTPC* promoter (2.2 kb; chr8:22,433,535–22,435,769; deposited at Addgene #201681). For the HA-SFTPC lines, HA-SFTPC-WT or I73T were cloned into pLenti-tetON-EF1a-tagRFP-2A-tet3G by Gibson assembly.

For CRISPRi, the organoids were initially transduced with doxycycline-inducible lentivirus containing dCas9 protein fused with 5' KRAB and 3' DHFR, and EF1a-TagRFP vector (Sun et al, 2021). Post-sorting, organoids were expanded before being transduced with lentivirus harboring U6 promoter-driven guide RNAs and EF1a-EGFP-CAAX, with the relevant guides (Ran et al, 2013) gRNA sequences for organoid CRISPRi are listed in Appendix Table S2. For activation of the CRISPRi system, 2 μg/ml dox and 10 nmol/l TMP were added to the medium for 5 days. For rescue experiments, dox and TMP were removed from the media for 14 days and organoids passaged where necessary before harvesting. CRISPRi experiments were carried out in three biologically independent lines.

### CRISPR screen

The forward genetic screen was undertaken using a protocol developed by the Lehner lab (Menzies et al, 2018) undertaken as a single experiment comprising numerous inbuilt features to ensure a representative readout (multiple guides per gene, >500-fold coverage, dual readout at two timepoints). A ubiquitome gRNA library consisting of approximately 1119 genes with 10 guides per gene was used (Menzies et al, 2018). The Cas9 activity of the HeLa-Cas9 line was confirmed by transducing with pKLV encoding β-2 micro-globulin gRNA and assessing loss of MHC class I from the plasma membrane by flow cytometry (with >80% loss considered acceptable). The amount of virus required for an MOI of 0.3 was determined by transducing $1 \times 10^6$ cells with varying amounts (25–400 μl) unconcentrated lentivirus and assessing BFP positivity by flow cytometry at 72 h post transduction. For the screen, 20 million cells were transduced to ensure 500-fold coverage and puromycin selection commenced at 48 h post transduction. After

5 days of selection, cells were harvested with 10 mM EDTA and stained using the SFTPC C-terminal domain antibody (Fig. EV4A). Cells most highly enriched for C-terminal antibody signal or GFP (~1%) were collected; half were harvested for genomic DNA extraction and half kept in culture for a further 7 days. This population underwent a further sort and the C-terminal antibody or GFP-enriched population was harvested. Genomic DNA was extracted using a Qiagen Puregene Core Kit. Lentiviral gRNA inserts were amplified in a two-step PCR reaction as previously described (Menzies et al, 2018), cleaned with AMPure XP magnetic beads and sequenced by MiniSeq. gRNA sequences for CRISPR hit validation are listed in Appendix Table S2.

### Flow cytometry

HeLa cells were detached using 10 mM EDTA and organoids dissociated to single cells before being washed with PBS and centrifuged at $500 \times g$ for 3 min to pellet. Non-specific staining was blocked with 10% FBS in PBS for 30 min then cells pelleted by centrifugation and incubated with primary antibody on ice for 30 min. Following two rounds of washing with PBS by centrifugation, cells were incubated with AF647-conjugated secondary antibodies for 30 min on ice, washed twice and filtered through 50-μm filters. Samples were analyzed on a Fortessa flow cytometer, and further analysis was undertaken using FlowJo software (10.0.0).

Flow cytometric sorting of single-cell dissociated organoids transduced with fluorescent markers was performed using a sorter (SH800S or BD FACSMelody) and analyzed using FlowJo.

### Immunoblotting

For immunoblotting, cells were subjected to triton lysis, SDS-PAGE electrophoresis, and immunoblotting as previously described (Malzer et al, 2013). Membranes were incubated overnight with primary antibody at 4 °C before washing and incubating with secondary antibody (1:20,000 IRDye® conjugated, various) for 1 h at room temperature. Membranes were visualized using a Li-Cor Odyssey imaging system.

### Immunofluorescence staining and microscopy

For immunostaining of HeLa cells, cells grown on coverslips were fixed with 4% PFA for 30 min, blocked with 10% FBS for 30 min then permeabilized for 30 min with 0.1% Triton. Cells were incubated with primary antibodies diluted in a blocking buffer overnight at 4 °C. After washing with PBS and incubated with Alexa-fluor conjugated secondary antibodies (1:500, various) for 1 h at room temperature before staining with DAPI (1 μg/ml) and mounting with Prolong Gold antifade.

For paraffin embedding (Fig. 4), organoids were seeded onto 0.4-μm transwell inserts embedded in 50% matrigel. Organoids were fixed with 4% PFA for 30 min before the membrane was released from the insert using a scalpel and embedded between layers of HistoGel before paraffin embedding. Samples were deparaffinised and rehydrated using sequential passes through xylene (x3) then ethanol (100%, 70%, 50%, 0%) then antigen retrieval undertaken by boiling slides in sodium citrate buffer (10 mM sodium citrate, 0.05% tween 20 pH 6.0) for 10 min. Samples were blocked using 1% BSA, 0.1% tween then incubated with primary antibodies overnight and Alexa-fluor conjugated secondary antibodies (1:500, various) for 1 h at room temperature before staining with DAPI, and mounting with Prolong Gold antifade. Imaging was undertaken using a Zeiss LSM880 with

airyscan and Zen black software. For HeLa live cell imaging, cells were imaged at 37 °C and in 5% $CO_2$.

For whole-mount immunostaining of organoids and 2D AT1-like cells (Figs. 1 and 2), the matrigel was completely removed from the cultured organoids using Cell Recovery Solution then fixed with 4% PFA for 30 min on ice. After rinsing in PBS washing solution containing 0.2% (v/v) Triton X-100 and 0.5% (w/v) BSA, the samples were incubated in permeabilization/blocking solution containing 0.2% (v/v) Triton X-100, 1% (w/v) BSA, and 5% normal donkey serum in PBS, overnight at 4 °C. Samples were then incubated with primary antibody at 4 °C overnight, washed and incubated with secondary antibody for 1 h at room temperature. Nuclei were counterstained using DAPI. Prior to the organoid imaging, step-wise treatments of 10%, 25%, 50%, and 97% (v/v) 2'–2'-thio-diethanol were followed for clearing. Images were taken using a Leica SP8 confocal microscope or Zeiss LSM880 with Airyscan and Zen black software.

For lysosomal fluorescence, organoids were incubated with Lysotracker green DND-26 at 37 °C for 2 h and immediately imaged under a fluorescence microscope.

For immunostaining of HEK-293T cells, cells grown on glass bottom microwell dishes coated with Poly-D-Lysine were washed with PBS and fixed with 4% PFA for 30 min. After washing with PBS the cells were blocked with 1% normal donkey serum for 15 min, washed with PBS and further blocked with 100% acetone for 5 min. After washing the cells with PBS-Tween-20 (0.05%) the cells were permeabilized for 10 min with 0.2% triton X. After washing with PBS-Tween-20, the cells were incubated with primary antibodies diluted in PBS-Tween-20 for 30 min at room temperature. After washing with PBS-Tween-20 and incubated with Alexa-fluor conjugated secondary antibodies (1:1000, various) for 30 min at room temperature before staining with DAPI (1 μg/ml) and mounting with VECTASHIELD® PLUS Antifade Mounting Medium. The specificity of primary and secondary antibodies were tested on coated dishes in the absence of cells. Images were taken using a Leica SP8 confocal microscope.

### Quantitative RT-PCR

Total RNA was isolated using an RNeasy kit including an optional DNase digestion step. Typically 500 ng RNA was used as the starting template to create cDNA using a high-capacity cDNA reverse transcription kit and heating samples to 25 °C for 10 min, 37 °C for 2 h and 85 °C for 5 min. RT-qPCR was undertaken in 96-well plates using 4.5 μl 1:10 cDNA and 10.5 μl of a master mix containing SYBR Green JumpStart Taq Readymix (Thermofisher, A46012) or SYBR Green PCR Master Mix (Applied Biosystems, 4309155). Primer sequences are listed in Appendix Table S1. Plates were run on a BioRad RT-PCR machine typically using the following program: 95 °C 2 min, 40× (95 °C for 30 s, 55 °C for 30 s, 72 °C for 30 s), 95 °C for 30 s.

### Bulk RNA sequencing

For fdAT2 organoid bulk RNA sequencing, RNA libraries of four biological lines of fdAT2 organoids at early (P1, 4, 6, 7) and late (P12, 13, 16, 17) passage (Appendix Table S3) were generated using a Qiagen RNeasy Kit including the optional DNAse digestion step. The quality of the RNA libraries was validated on Agilent 2200 Tapestation before sequencing on an Illumina NovaSeq 6000 at Novogene. A comparison between the RNA sequencing data of fdAT2 organoids and publicly available data of fetal organoid types and other alveolar cell types was performed: fetal early tip progenitor organoids GSM5393370 and GSM5393371; fetal late tip progenitor organoids GSM5393372 and GSM5393373 (Lim et al, 2023); PSC-iAT2s, GSM5578511, GSM5578512 and GSM5578513 (Abo et al, 2022); cultured adult AT2 cells GSM5578508, GSM5578509, and GSM5578510 (Abo et al, 2022); freshly isolated adult AT2 cells GSM2537127, GSM2537128 and GSM2537129 (Jacob et al, 2017). The raw RNA sequencing data was run by a bioinformatics pipeline, nf-core/rnaseq (Ewels et al, 2020). A list of differentially expressed genes was extracted using the counted reads and R package edgeR version 3.40.2 (Robinson et al, 2010). GO biological processes term enrichment, KEGG pathway, and gene set enrichment analysis were performed using DAVID (Huang et al, 2009) and R package fgsea (Korotkevich et al, 2021) packages.

### Organoid single-cell RNA sequencing

Four biological replicates of the fdAT2 organoids (passage numbers 11, 15, 15, 16; Appendix Table S3) were harvested individually and dissociated into single cells using TrypLE Express enzyme (Thermo Fisher). The cell suspension was pooled, passed through a 45-μm filter, pelleted, resuspended in 1 ml of 0.04% BSA/PBS, and counted. In total, 5000 cells per donor were pooled and single cell RNA-seq was carried out according to 10x Chromium Single Cell 5' Kits (v1). Library generation for 10x Genomics analysis was performed following the Chromium Single Cell 5′ Reagents Kits and sequenced on a NovaSeq 6000 S4 Flowcell (paired-end (PE), 150-bp reads) to achieve an average of 80,000 PE reads per cell. scRNA-seq data were mapped with STARsolo 2.7.10a (Kaminow et al, 2021) using the GRCh38 reference distributed by 10X (3, 0, 0, derived fromEnsembl 93). Cells were called by a customized EmptyDrops (Lun et al, 2019) method from CellRanger 3.0.2 (distributed as emptydrops on PyPi). The SNPs of the four donors were inferred from the bulk RNA-seq data using cellsnp-lite (--minMAF 0.1 –minCOUNT20) (Huang and Huang, 2021). These SNPs were used to demultiplex the scRNA-seq data into four donors using souporcell (v2.5, -k 4 –known_genotypes $VCF) (Heaton et al, 2020).

Organoid scRNA-seq data were initially manually annotated based on the expression of marker genes (in Fig. 1). For the prediction of organoid cell types from previously published Human Lung Cell Atlas data (in Fig. EV2), we used specific human adult (Madissoon et al, 2023) and fetal (He et al, 2022) atlases as reference datasets. We randomly selected 20,000 cells based on cell type from the human adult lung cell atlas (Madissoon et al, 2023) for subsequent analysis. We first used the Seurat package (v4.0) for data integration, followed by the TransferData function to predict the cell types in the organoids.

### Electron microscopy

Whole organoids were fixed with 2% PFA, 2.5% glutaraldehyde, 0.1 M cacodylate buffer, pH 7.4. Organoids were secondarily fixed with 1% osmium tetroxide/1.5% potassium ferrocyanide and then incubated with 1% tannic acid in 0.1 M cacodylate buffer to enhance membrane contrast. Organoids were washed with water before being dehydrated using increasing percentages of ethanol (70,%, 90%, 100%). Samples were embedded in beam capsules in CY212 Epoxy resin and resin cured overnight at 65ºC. Ultrathin sections were cut using a diamond knife mounted to a Reichart

ultracut S ultramicrotome. Sections were collected onto piloform-coated slot grids and stained using lead citrate. Sections were viewed on a FEI Tecnai transmission electron microscope at a working voltage of 80 kV.

### Bioinformatic and statistical analysis

Statistical analysis of CRISPR screen sequencing data was performed using MAGeK (Li et al, 2014). Of note, the calculated significance of a gene is not necessarily directly proportional to its biological significance; relative gRNA efficiency and lethal phenotypes generated from knockdowns may preclude the enrichment of other genes of functional relevance.

Data expressed as mean ± standard deviation (SD) from at least three independent experiments. Each statistical test is described in the figure legends. GraphPad Prism software (version 9.5.1) was used for statistical analysis and data visualization.

## Data availability

No new code was generated for the bulk RNA sequencing; any additional information required to re-run the code and repeat the analyses reported can be requested from the corresponding authors. Bulk RNA sequencing data have been deposited at GEO: GSE237359. Single-cell RNA sequencing codes were deposited at https://github.com/brianpenghe/2024_AT2_organoids. Single-cell sequencing data have been deposited at GEO: GSE237359.

The source data of this paper are collected in the following database record: biostudies:S-SCDT-10_1038-S44318-024-00328-6.

## Peer review information

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

## Acknowledgements

KL is supported by the Korea University Grant and by the National Research Foundation of Korea (NRF) grant funded by Korea government (MSIT) (RS-2024-00342036). ENR and JAD are supported by an MRC Clinician Scientist Fellowship (MR/S005552/1). PH, SAT and JCM are supported by the Chan Zuckerberg Initiative foundation (CZIF2022-007488). PH holds a non-stipendiary research fellowship at St Edmund's College, University of Cambridge. DS is supported by a Wellcome Trust PhD studentship (109146/Z/15/Z) and the Department of Pathology, University of Cambridge. JRE is supported by a Sir Henry Dale Fellowship jointly funded by the Wellcome Trust and the Royal Society (216370/Z/19/Z). LEM was supported in part by the National Institute of General Medical Sciences of the National Institutes of Health under Award Number P20GM103499. J-HL and JHB are supported by a Wellcome Senior Research Fellowship (221857/Z/20/Z) and the Suh Kyungbae Foundation (SUHF-20010033). SJM is supported by the MRC (MCMB MR/V028669/1 and MR/R009120/1), EPSRC (EP/R03558X/1), Cambridge Biomedical Research Centre (BRC-1215-20014); Asthma+Lung UK (ALUK), and the Victor Philip Dahdaleh Foundation. ELR and LD are supported by the Medical Research Council (MR/P009581/1) and Wellcome Discovery Award (225221/Z/22/Z). PJL and DJHVB are supported by a Wellcome Trust Principal Research Fellowship (210688/Z/18/Z) and an MRC project grant (MR/V011561/1). We acknowledge core funding to the Gurdon Institute from the Wellcome Trust (203144/Z/16/Z) and CRUK (C6946/A24843). This research was also supported by the CIMR Flow Cytometry Core Facility.

## Author contributions

**Kyungtae Lim**: Conceptualization; Software; Formal analysis; Validation; Investigation; Methodology; Writing—original draft. **Eimear N Rutherford**: Conceptualization; Software; Formal analysis; Validation; Investigation; Methodology; Writing—original draft. **Livia Delpiano**: Validation; Investigation; Methodology. **Peng He**: Validation; Investigation; Methodology. **Weimin Lin**: Validation; Investigation; Methodology. **Dawei Sun**: Validation; Investigation; Methodology. **Dick JH Van den Boomen**: Software; Formal analysis; Supervision; Validation; Investigation; Methodology. **James R Edgar**: Resources; Validation; Investigation; Methodology. **Jae Hak Bang**: Validation; Investigation; Methodology. **Alexander Predeus**: Validation; Investigation; Methodology. **Sarah A Teichmann**: Resources. **John C Marioni**: Resources. **Lydia E Matesic**: Resources; Validation; Investigation; Methodology. **Joo-Hyeon Lee**: Resources; Supervision; Funding acquisition; Writing—review and editing. **Paul J Lehner**: Conceptualization; Resources; Supervision; Funding acquisition; Writing—review and editing. **Stefan J Marciniak**: Conceptualization; Resources; Supervision; Funding acquisition; Methodology; Writing—review and editing. **Emma L Rawlins**: Conceptualization; Supervision; Funding acquisition; Writing—review and editing. **Jennifer A Dickens**: Conceptualization; Supervision; Validation; Investigation; Methodology; Project administration; Writing—review and editing.

Source data underlying figure panels in this paper may have individual authorship assigned. Where available, figure panel/source data authorship is listed in the following database record: biostudies:S-SCDT-10_1038-S44318-024-00328-6.

## Disclosure and competing interests statement

SAT is a remunerated member of the scientific advisory boards of Qiagen, Foresite Labs, and Element Biosciences, a co-founder and equity holder of TransitionBio, and a part-time employee of GlaxoSmithKline since January 2024. JCM has been an employee of Genentech, Inc. since September 2022. The remaining authors declare no competing interests.

# Expanded View Figures

**Figure EV1.   Characterization of AT2 organoids.**                                                                                    ▶

(A) Derivation and establishment of fdAT2 organoids from lung tip progenitor cells (upper panel), or proximal airway progenitors (lower panel) of human fetal lungs at 22 pcw. Upper panel: The isolated tip progenitor cells were immediately transduced and selected based on *SFTPC*-GFP and EF1a-TagRFP after 48 h of transduction; *SFTPC*-GFP and EF1a-TagRFP reporter positive cells were efficiently expanded into fdAT2 organoids when grown in AT2 medium for 3 weeks. Lower panel: Proximal airway cells were immediately transduced with *SCGB3A2*-GFP, EF1a-TagRFP reporter lentivirus and the airway progenitor cells were selectively isolated by *SCGB3A2*-GFP after 48 h of transduction; *SCGB3A2*-GFP, *EF1a*-TagRFP reporter positive cells and expanded into small AT2-like organoids when grown in AT2 medium for 3 weeks, but efficiently formed airway organoids when grown in airway medium for the same period. Scale bar, 50 μm. (B) Size of organoids expanded from tip progenitors (*SFTPC*-GFP$^+$) or proximal airway progenitors (*SCGB3A2*-GFP$^+$) in AT2 medium was measured; mean ± SD, $n = 50$ organoids from one biological line (unpaired t test, two-tailed; $n = 1$ for each group); $P = 0.000000000000297$. (C) Expression of mature SFTPC protein in organoids expanded from tip progenitors or airway progenitors in AT2 medium. DAPI, nuclei. Scale bar, 50 μm. (D) Cultured fdAT2 organoids at early and late passages, stably expressing *SFTPC* promoter-driven GFP (*SFTPC*-GFP). Two independent lines of AT2 organoids at P3 and P17, and P4 and P20. Scale bar, 50 μm. (E) RT-qPCR analysis of alveolar type 2 cell lineage markers, *NKX2.1*, *SFTPC*, *ABCA3*, and *LAMP3*, in 7–9 pcw and 16–22 pcw tip progenitor organoids, and fdAT2 organoids at P12, P20, and P21. Data were normalized to 7–9 pcw tip organoids; mean ± SD, $n = 3$ biologically independent organoid lines (one-way ANOVA with Tukey multiple comparison post-test). (F) Additional electron microscopy showing the presence of lamellar bodies with characteristic concentric lamellar membranes within the cytosol. Scale bar, 1 μm. (G) Immunoblot of mature forms of SFTPC and SFTPB in human fetal lung tip progenitor-derived organoids at 7–9 pcw and 16–22 pcw, respectively, and the fdAT2 organoids. Two biologically independent organoid lines were used. (H) FdAT2 organoids were immunostained for E-cadherin (cyan) and mature SFTPC (red). Mature SFTPC is observed within the cells packaged into lamellar bodies and secreted into the lumen of the organoids. The lumen border is labeled by the dotted line. Arrowheads indicate cells without mature SFTPC expression on the merged panel. Scale bar, 50 μm. (I) FdAT2 organoids were cultured in AT2 medium for 2 weeks, in the presence (control) or absence of FGF7 (-FGF7) and AT2 lineage markers were measured by RT-qPCR (J) after 7 and 14 days of culture. Data were normalized to AT2 organoids cultured in the AT2 medium containing FGF7 (control); mean ± SD, $n = 3$ biologically independent organoid lines (one-way ANOVA with Tukey multiple comparison post-test). (K) Mature SFTPC protein expression was visualized with a proliferation marker, KI67, and E-cadherin by immunofluorescence staining, at 14 days of culture. DAPI, nuclei. Scale bar, 50 μm. Source data are available online for this figure.

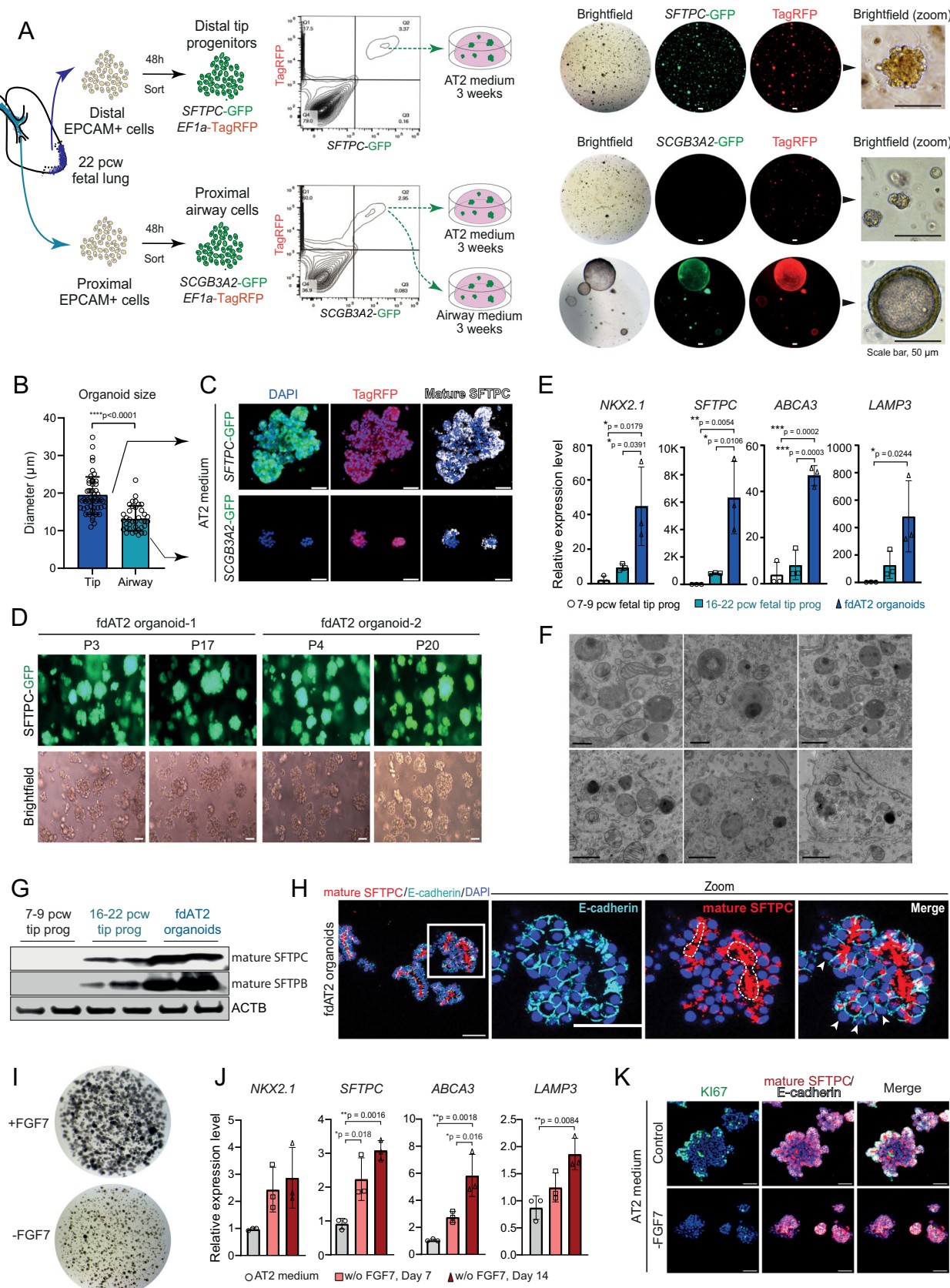

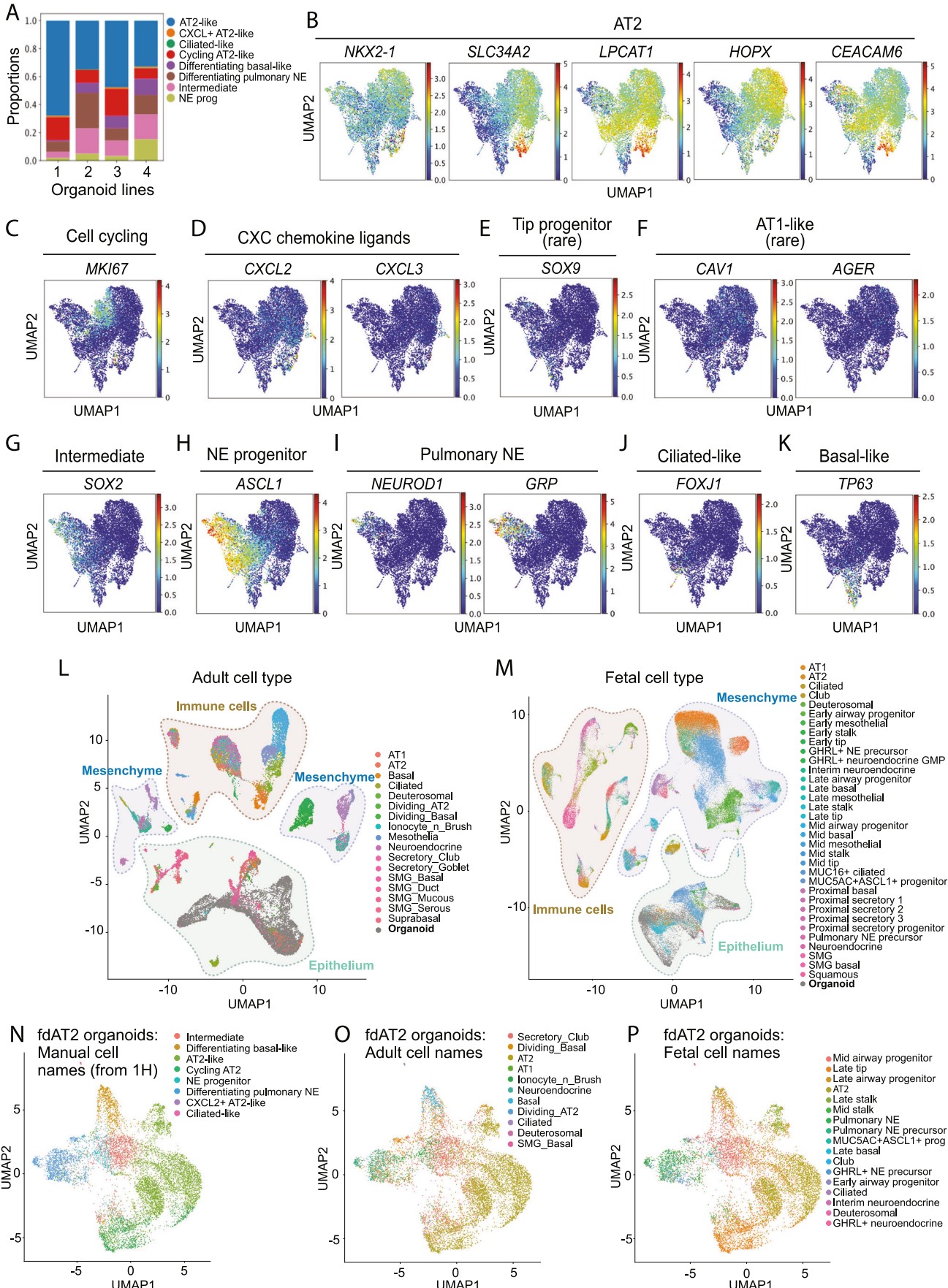

◄   **Figure EV2.   Single-cell RNA sequencing analysis of the fdAT2 organoids.**

(**A**) Graph showing the proportions of each cell type obtained from the four independent organoid lines. (**B–K**) UMAP plots showing transcript expression of AT2 lineage markers *NKX2-1, SLC34A2, LPCAT1, HOPX, CEACAM6* (**B**), *MKI67* (**C**), CXC chemokine ligands *CXCL2, CXCL3* (**D**), *SOX9* (**E**), AT1-like markers *CAV1, AGER* (**F**), *SOX2* (**G**), *ASCL1* (**H**), neuroendocrine cells *NEUROD1, GRP* (**I**), *FOXJ1* (**J**), *TP63* (**K**). (**L, M**) UMAPs showing fdAT2 organoid scRNA-seq data integrated with published adult (**L**) or fetal (**M**) scRNA-seq atlas data. FdAT2 organoid cells are shown in gray. (**N, O**) UMAPs showing fdAT2 organoid scRNA-seq with cell clusters named by manual annotation as in Fig. 1H (**N**), label transfer from the adult cell atlas (**O**), or label transfer from the fetal cell atlas (**P**).

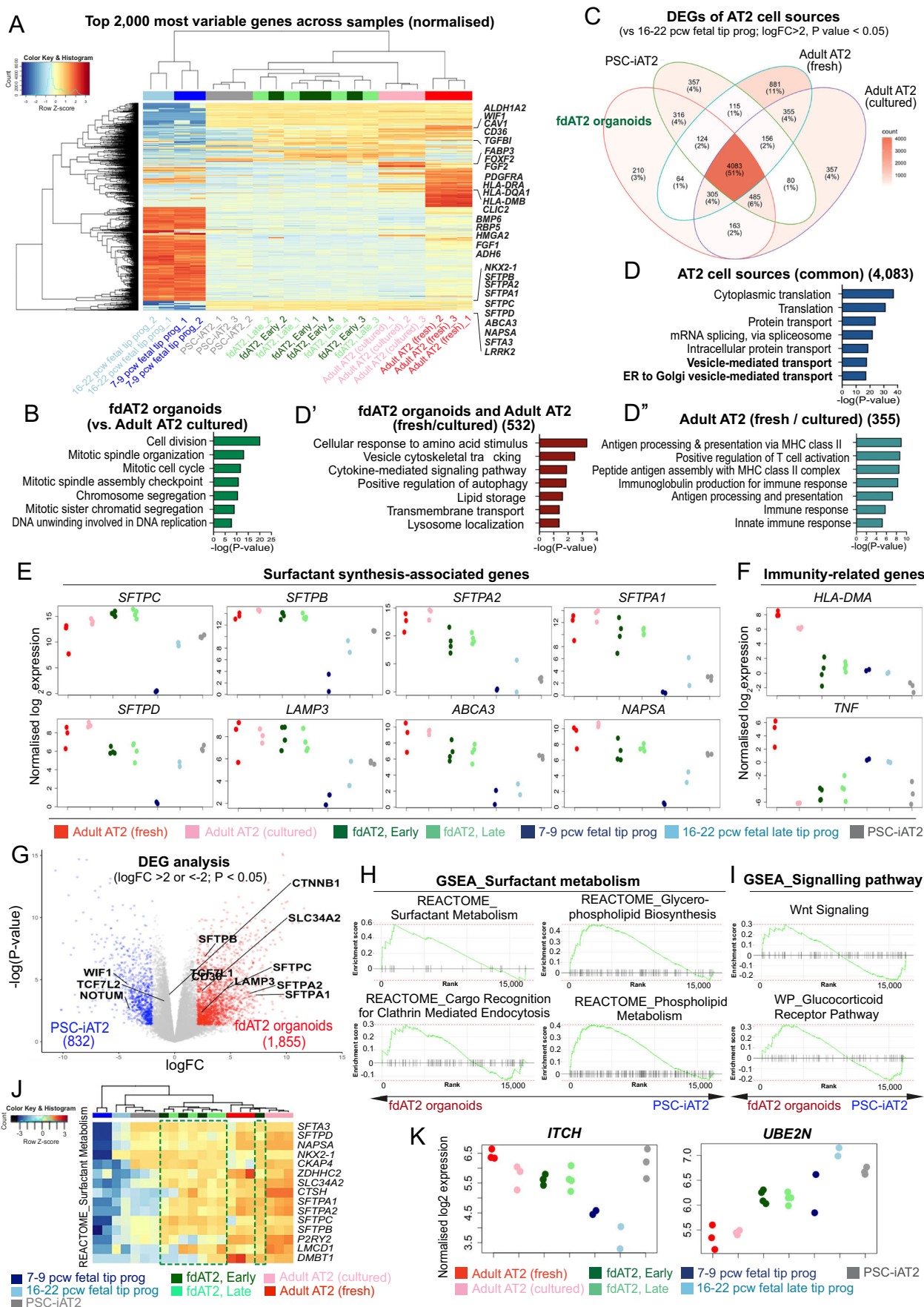

**A** Top 2,000 most variable genes across samples (normalised)

**B** fdAT2 organoids (vs. Adult AT2 cultured)

**C** DEGs of AT2 cell sources (vs 16-22 pcw fetal tip prog; logFC>2, P value < 0.05)

**D** AT2 cell sources (common) (4,083)

**D'** fdAT2 organoids and Adult AT2 (fresh/cultured) (532)

**D"** Adult AT2 (fresh / cultured) (355)

**E** Surfactant synthesis-associated genes

**F** Immunity-related genes

**G** DEG analysis (logFC >2 or <-2; P < 0.05)

**H** GSEA_Surfactant metabolism

**I** GSEA_Signalling pathway

**J** (heatmap)

**K** ITCH / UBE2N

◄  **Figure EV3.  Comparative transcriptomic analysis of fetal-derived AT2 organoids with other AT2 sources.**

(A) Heatmap analysis of the top 2000 most variable genes across all samples. (B) GO analysis of genes highly enriched in the fdAT2 organoids compared to the cultured adult AT2 cells. (C) Venn diagram illustrating the number and the proportion of unique or shared genes from AT2 cells of different sources. The genes for each AT2 cell type that were differentially expressed compared to fetal 16–22 pcw tip progenitor organoids were included ($\log_2$FC > 2, *P* value < 0.05; Dataset EV2). (D, blue) GO analysis of DEGs shared between AT2 cells of different origin, including fdAT2 organoids, PSC-iAT2, and cultured and freshly isolated adult AT2 cells; related to Fig. 1L. 4083 genes commonly shared by all AT2 fate cell types. (D′, red) genes shared by fdAT2 organoids and cultured and/or freshly isolated adult AT2 cells (D″, cyan) genes shared by cultured and freshly isolated adult AT2 cells. All GO analysis was performed using DEGs with $\log_2$FC > 2, *P* value < 0.05; *P* values for each pairwise comparison are shown in Datasets EV1–3. (E, F) Expression levels of surfactant synthesis-associated genes (E) and immunity-related genes (F) across the different AT2 cell sources. (G) Volcano plot describing the direct comparison of fdAT2 organoids and PSC-iAT2. 1855 and 832 genes were differentially enriched in AT2 organoids and PSC-iAT2, respectively (logFC > 2, P value < 0.05; related to Dataset EV3). (H, I) Gene set enrichment analysis (GSEA) of surfactant metabolism and signaling pathway-associated gene sets between fdAT2 organoids and PSC-iAT2. (J) Heatmap of a gene set associated with surfactant metabolism from REACTOME. Green box, fdAT2 organoids. (K) Relative expression of E3 ligases *ITCH* and *UBE2N* in AT2 cells and tip progenitor organoids.

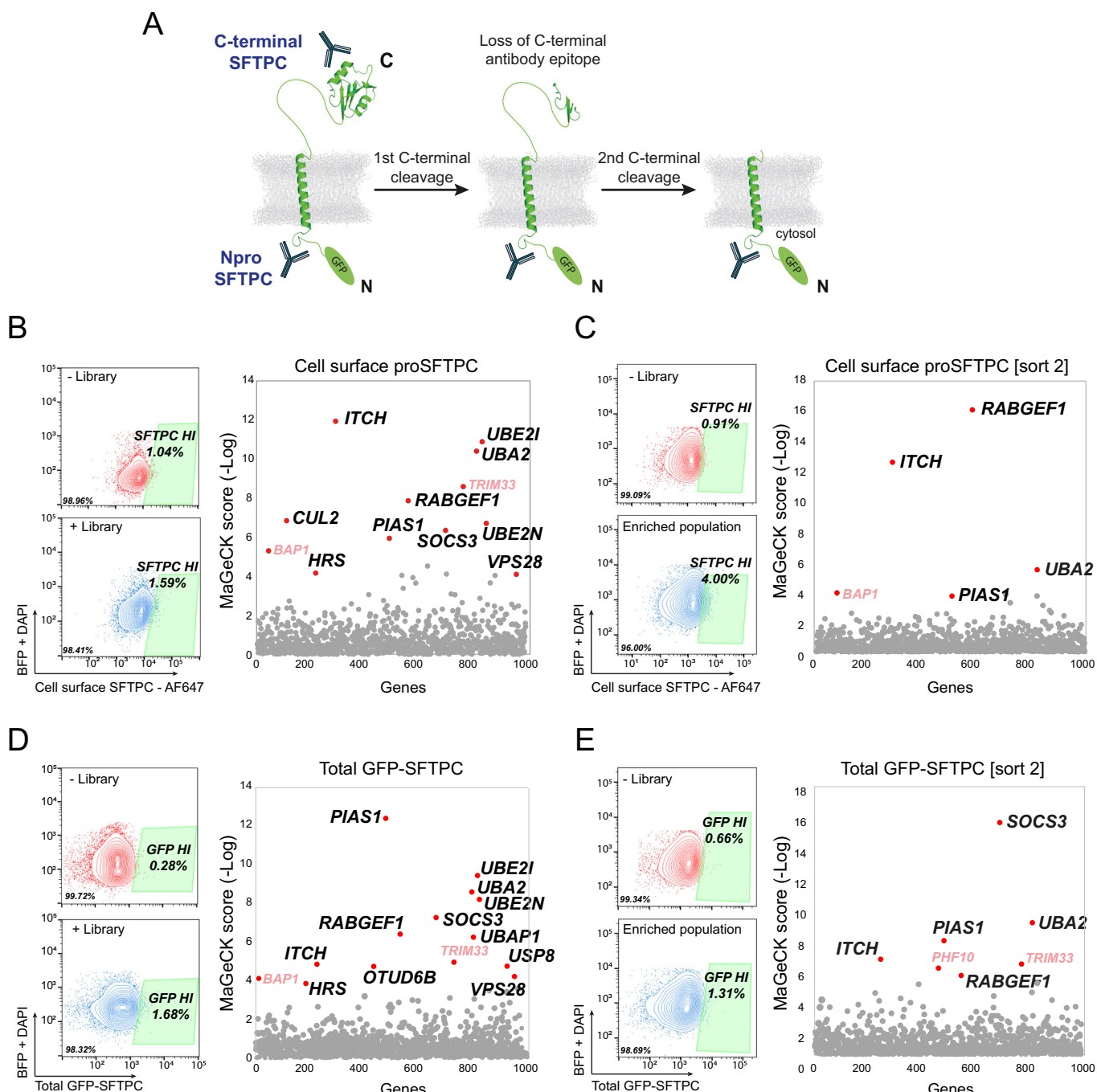

**Figure EV4.   A forward genetic screen identifies candidate proteins involved in SFTPC processing and trafficking.**

(A) Schematic showing location of GFP tag and epitopes for N-terminal and C-terminal SFTPC antibodies. Only full-length proSFTPC is recognized by the C-terminal antibody used for flow cytometry assays. (B–E) Flow cytometry gating strategy and MAGeCK relative enrichment scores for genes whose depletion results in increased cell surface SFTPC (B, day 7 and C, day 14) or increased total eGFP-SFTPC (D, day 7 and E, day 14) post transduction with ubiquitome sgRNA library. Genes highlighted red (BAP1, TRIM33 and PHF10) are commonly enriched but non-specific transcription-related hits from forward genetic screens.

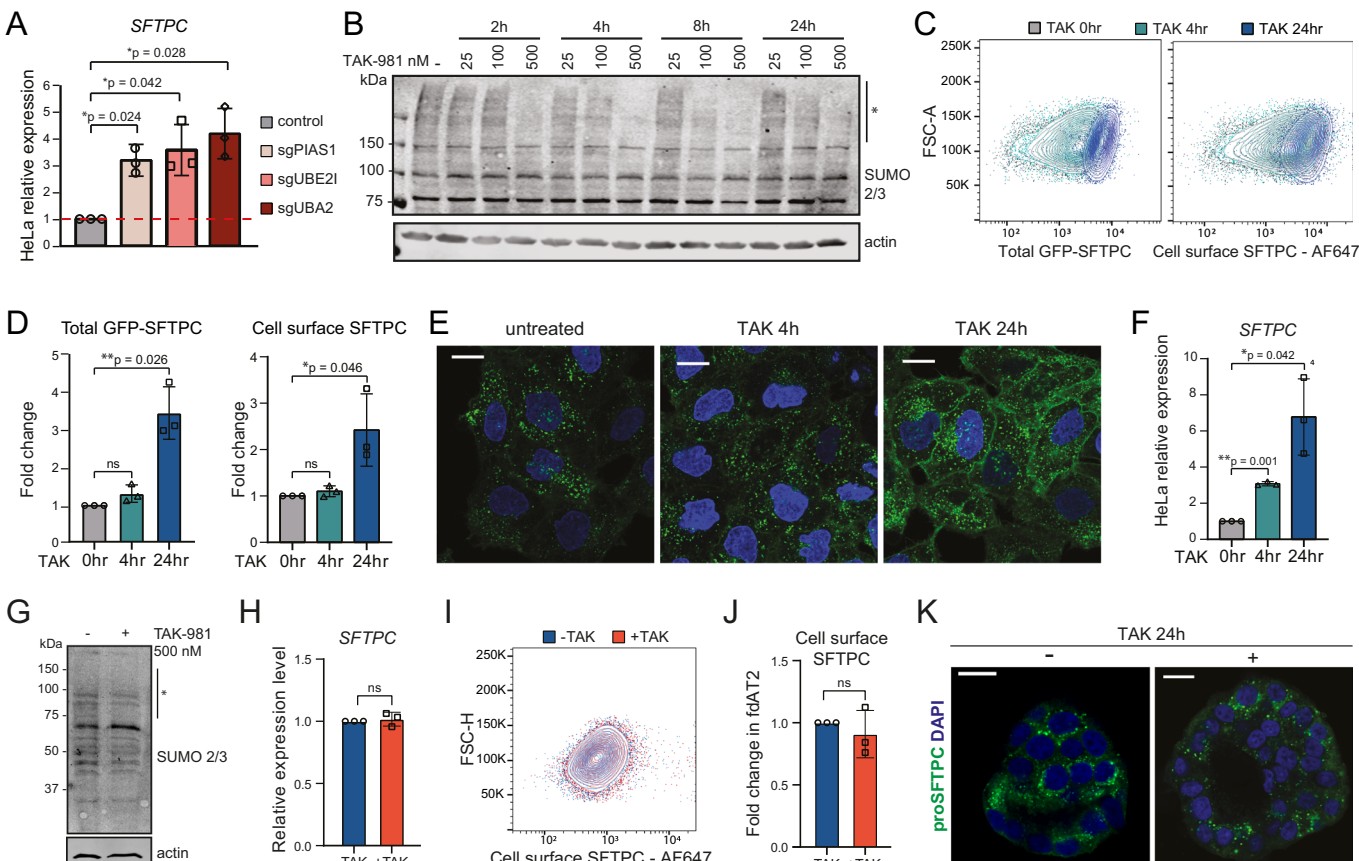

**Figure EV5. SUMOylation is not involved in the expression or maturation of SFTPC when expressed from an endogenous promoter.**

(A) Relative expression of SFTPC mRNA in GFP-SFTPC-Cas9 control cells and UBE2I, UBA2, and PIAS1 knockout pool; mean ± SD, $n = 3$ independent repeats (one-way ANOVA with Tukey multiple comparison post-test). (B) SUMO 2/3 immunoblot of TAK-981-treated HeLa cells (SUMOylated proteins are seen as a smear of higher molecular weight species (*) HeLa cells treated with 500 nM TAK-981 were assessed for total and cell surface full-length SFTPC as measured by flow cytometry (C, D) and live cell confocal microscopy (E), and for relative expression of SFTPC mRNA (F); mean ± SD, $n = 3$ independent repeats (one-way ANOVA with Tukey multiple comparison post-test). Scale bar, 20 μm. (G) SUMO 2/3 immunoblot of fdAT2 treated with 500 nM TAK-981 for 24 h. * = Smear of sumoylated proteins. fdAT2 were treated with 500 nM TAK-981 for 24 h, assessed for relative expression of SFTPC mRNA (H), total and cell surface SFTPC as measured by flow cytometry (I, J) and microscopy (K); mean ± SD $n = 3$ biologically independent organoid lines (one-way ANOVA with Tukey multiple comparison post-test). Scale bar, 10 μm. Source data are available online for this figure.

