## [Peer Review File · The EMBO Journal]

Human fetal lung-derived alveolar organoids reveal mechanisms of surfactant maturation

Kyungtae Lim, Eimear Rutherford, Livia Delpiano, Peng He, Weimin Lin, Dawei Sun, Dick Van den Boomen, James Edgar, Jae Bang, Alexander Predeus, Sarah Teichmann, John Marioni, Lydia Matesic, Joo-Hyeon Lee, Paul Lehner, Stefan Marciniak, Emma Rawlins, and Jenny Dickens

Corresponding authors: Emma Rawlins (elr21@cam.ac.uk) , Jenny Dickens (jac72@cam.ac.uk)

Review Timeline:

Submission Date:	11th Sep 24
Editorial Decision:	7th Oct 24
Revision Received:	1st Nov 24
Accepted:	18th Nov 24

Editor: Ieva Gailite

Transaction Report:

(Note: This manuscript transferred to The EMBO Journal following peer review at another journal. The peer review comments and authors' responses were made available as agreed with the authors and the other journal, and were taken into account for the decision process at The EMBO Journal. With the exception of the correction of typographical or spelling errors that could be a source of ambiguity, letters and reports are not edited. Depending on transfer agreements, referee reports obtained elsewhere may or may not be included in this compilation. Referee reports are anonymous unless the Referee chooses to sign their reports.)

We respectfully disagree with reviewer 3 about the quality of our data, and about the value of the experiments and analyses that they suggested. Please see point-by-point rebuttal below, concentrating on points 2-4 as per guidance from Dr. Stelios Lefkopoulos (editor, Nature Cell Biology). Reviewers' comments are shown in blue for clarity of reading.

1. Staining for HT2-280 ((Extended Data Fig. 2G): The assumption in interpreting these data is that the blue population is the negative control (and it is unclear which negative control exactly), the red the stained cells and that the dot plots are overlays, as in most flow software different colors indicate different samples. If so, then the same negative control was used for three independent samples, since the blue populations are identical. Furthermore, the data would then suggest that the cells are 100% positive for HT2-280. This seems unlikely given the scRNAseq data. HT2-280 may stain some more immature cells, it does not stain all lung cells, by far. Also, HT2-280 is unconjugated, which complicates dual staining for flow. How exactly was the co-staining with EPCAM done? The flow plots are not present in the source data files.

However, in their rebuttal, the authors state that some negative control staining was omitted, hence all contours may be from the same sample. In that case, it is difficult to explain how the blue populations are identical, and what the different colors mean. And in that case, only a minority of the cells are AT2. Percentages are not given either.

HTII-280 immunoreactivity is extremely widespread in fetal development (PMID: 28665271, 36493780) and the function of this membrane protein is unknown. As these are fetal lung-derived organoids, the staining for HTII-280 provided by IF (and additionally now by flow cytometry on persisting request by reviewer 3) does not provide any meaningfully interpretable result when inferring the identity of cells in fdAT2 organoids. The IF in Ext Data 2E shows that HTII-280 is expressed at some level in all the cells, which is in agreement with the flow cytometry in Ext Data 2G.

For clarity, the negative control in Extended Data Fig. 2G was fdAT2 stained only with secondary antibody; this has been clarified in the figure legend. The negative control data removed was unrelated to the HTII-280 staining; it was simply negative control staining of HEK cells by IF that was previously included after the first round of revisions but, in our view, was insufficiently important to retain in the manuscript with the increasing space constraints following further revision.

2. The additional scRNAseq analysis. Integration indeed shows that the cells cluster near lung epithelial cells. However, it is difficult to evaluate cell identity from the plots shown (Extended Data Fig. 3L, M). In fact, most cells do not overlap with the AT2 cells in these analyses. These plots therefore raise even more questions about the identity of the cells, although I would agree that interpretation of integrated data is perhaps not straightforward in this case. The label transfer analysis in panels N, O and P however show that not even the majority of the cells are AT2-like, let alone AT2 cells. A very substantial fraction are neuroendocrine, others have airway phenotypes. In the end, it seems that these are very heterogeneous fetal lung organoids. How this can be reconciled with the IF in the manuscript that shows uniform expression of AT2 markers throughout (despite claims from the authors to the contrary, but this is very hard to see in the images) and with likely 100% HT2-280 positivity by flow is difficult to understand. It would also perhaps be useful to look for some other markers that were identified in scRNAseq by IF, so the reader has a good idea of what the organoids really contain.

There are 2 issues in this comment:

a. Perceived organoid heterogeneity:

Between revision rounds, reviewer 3 has changed their view from having no concern that there are some airway cells in the organoids to concluding that we are growing a highly heterogeneous mixed population. This is despite being presented with essentially the same data. This opinion appears to be based on one single cell sequencing experiment, which the reviewer acknowledges is difficult to interpret, and overlooks all the other data in the manuscript.

We are aware that it is the single cell sequencing experiment which is the outlier of our overall dataset, and we are prevented by cost and time constraints from repeating it. Some organoid lines for scRNA-seq were grown slightly longer than normal without passaging and we now understand they likely became somewhat hypoxic. The work of a PhD student in the Rawlins lab has since shown that three types of lung organoids (foetal multipotent progenitors, fdAT2 used in this manuscript under consideration, and adult derived AT2s) spontaneously differentiate to airway fate, particularly basal and neuroendocrine cells, in hypoxic culture, or when they become functionally hypoxic (e.g. longer periods between passage/over-crowding) (doi: <https://doi.org/10.1101/2024.08.09.607336>). It can be seen very clearly from Fig 1H,J that the basal and neuroendocrine cells in the cultures express a gradient of *SFTPC* along their clusters and are most likely derived from the fdAT2s. These observations are highly consistent with the preprint from the Blackwell and Kropski labs which shows that adult AT2 cells also undergo HIF-mediated airway differentiation in organoid culture (doi: <https://doi.org/10.1101/2023.09.17.557477>).

All of the other data in the manuscript supports our conclusion that when the organoids are passaged weekly (and therefore prevented from becoming overcrowded and hypoxic) they retain a largely AT2 phenotype.

b. Mapping of organoid cells to the *in vivo* cell atlases:

The reviewer is uncomfortable that the AT2 cells from the fdAT2 organoids do not overlap perfectly with the *in vivo*-derived cells in UMAP space after the two data sets were integrated, though acknowledges this was not straightforward. We could have spent a lot of time altering the parameters in Seurat to ensure that the *in vivo* and *in vitro* clusters overlapped on the UMAP. However, we are growing cells in a highly artificial system in Matrigel and it would be extremely surprising if they overlapped perfectly with their *in vivo* counterparts. The label transfer process clearly shows that based on the scRNA-seq data, the closest match for our cells *in vivo* is AT2. This is also supported by our functional assays, which demonstrate the synthesis, trafficking, and secretion of mature surfactant proteins in fdAT2 cells. Moreover, Fig. 1L (red and pink dots, reanalysed publicly available data from the Kotton lab) shows clearly that even *in vivo*-derived fresh and cultured AT2s from the same donor are not transcriptionally identical.

3. I asked previously to show flow cytometry for surface SFTPC (Fig. 1E, 4E, 4I) in conjunction with an 'empty' fluorescence channel to make sure that we are not looking at aspecific fluorescence, which would appear on the diagonal in such a dot plot. The authors however show side scatter or forward scatter in the manuscript, although there is a correct plot in the rebuttal. Can all these panels be modified accordingly? Plotting against side or forward scatter does not rule out aspecific fluorescence in any way. I also want to point out

that in the source data, the plot for Fig. 1E, but with colors reversed, shows forward scatter on the Y-axis. Fig. 1E show side scatter.

Reviewer 3 repeatedly had the rather unusual request of presenting our flow data in figure 1 using aspecific fluorescence on the y-axis. During a previous round of rebuttal we demonstrated no aspecific fluorescence shift for the figure in question (included again below, for clarity). In the final comments from this reviewer, a further request was made for applying this style of presentation to Figure 4. This is not possible for figure 4; all 4 channels were used during the flow cytometry (blue and green: sgRNA, red: stably expressed with KRAB-dCas9-DHFR, far red: SFTPC staining) so we cannot show aspecific fluorescence from an unused channel.

The discrepancy between the source data and Fig.1E FSC/SSC was an oversight for which we apologise and has been amended.

4. Another issue that comes back is the fact that the organoids can be maintained for 20 passages. This is used as the justification for this approach compared to adult AT2 organoids. However, there are still no data showing that, despite my request in the first review. Cellular expansion curves and data on cloning efficiency would be very helpful here.

This comment from reviewer 3 after the second round of revisions is the first time that we have been asked for growth curves and cloning efficiency of the fdAT2 cells, so we do not have the data. If we started now, it would take at least 6 months to produce these data. In the first round of revisions, we deliberately worked with high passage number organoids (see Extended Data Fig. 2 in which the organoids were passage 19 and 20). Moreover, Fig. 4 shows that these organoids are robust to two rounds of lentiviral transduction and subsequent expansion for cell sorting. The statement about 20 passages in the manuscript just says that we have split organoids weekly for 20 passages and that they retain expression of surfactant-

related genes by IF (shown in extended data Fig. 2). We maintain that this is fair given all the data presented.

Dear Emma,

Thank you for submitting your manuscript to The EMBO Journal. I have now received input from an arbitrating advisor, who finds your response to the remaining referee concerns from the previous assessment round satisfactory. Therefore, there now remain only a number of editorial points and formatting steps that need to be completed before I can extend official acceptance of the manuscript:

1. Please submit up to five keywords.
2. Please check that the funding information is correct and identical both in the manuscript and our online system. Currently, the Gurdon Institute from the Wellcome Trust (203144/Z/16/Z) and CRUK (C6946/A24843); the CIMR Flow Cytometry Core Facility are missing from our online system.
3. Please submit a complete author checklist, which you can download from our author guidelines (<https://www.embopress.org/pb-assets/embo-site/EMBO%20Press%20Author%20Checklist-1642513524327.xlsx>). Please insert information in the checklist that is also reflected in the manuscript. The completed author checklist will also be part of the Review Process File.
4. Please make sure that the order of the sections in the manuscript is as follows: abstract, introduction, results, discussion, materials & methods, data availability section, acknowledgments, disclosure statement and competing interests, references, main figure legends, tables, expanded figure legends.
5. All Materials and Methods need to be described in the main text using our 'Structured Methods' format. According to this format, the Methods section includes a Reagents and Tools Table (listing key reagents, experimental models, software and relevant equipment and including their sources and relevant identifiers) followed by a Methods and Protocols section describing the methods, ideally using a step-by-step protocol format. The aim is to facilitate adoption of the methodologies across labs. Please download and fill our Reagents and Tools Table template (.docx), which you can find in our author guidelines: <https://www.embopress.org/page/journal/14602075/authorguide#structuredmethods>
An example of a Method paper with Structured Methods can be found here: <https://www.embopress.org/doi/10.15252/msb.20178071>.
When submitting your revised manuscript, please upload it as a separate file choosing the file type "Reagent Table". The information currently provided Supplementary Table 1 could be adapted to this format.
6. Please rename "Competing Interests" section into "Disclosure and competing interests statement".
7. We require a Data Availability Section at the end of Materials and Methods, which should include links to primary datasets produced in this study. Further information can be found at <https://www.embopress.org/page/journal/14602075/authorguide#dataavailability>
8. At EMBO Press we ask authors to provide source data for the main and EV figures. Our source data coordinator Hannah Sonntag will contact you to discuss which figure panels we would need source data for and will also provide you with helpful tips on how to upload and organize the files.
9. CRediT has replaced the traditional author contributions section because it offers a systematic, machine-readable author contributions format that allows for more effective research assessment. Please remove the Authors Contributions from the manuscript and use the free text boxes beneath each contributing author's name in our online submission system to add specific details on the author's contribution. More information is available in our guide to authors.
10. Please assemble references into a single list and update according to The EMBO Journal style - where there are more than 10 authors on a paper, the first 10 should be listed, followed by 'et al.' Please see further information here: <https://www.embopress.org/page/journal/14602075/authorguide#referencesformat>
11. In addition to the main figures, we can accommodate up to five Expanded View (EV) figures. Please assemble the rest of the EV figures in a single Appendix pdf file, which should be prefaced by a short table of contents including page numbers. The nomenclature should be updated to that of Figure EV1-5 and Appendix Figure S1 etc, throughout the manuscript and Appendix file.
12. There is a reference to "data not shown" in line 290. According to our policy, which does not permit references to "data not shown", please include this information in the Appendix. Please see also <https://www.embopress.org/page/journal/14602075/authorguide#unpublisheddata>.
13. During our routine image quality check, we noted that Extended Data Figures 2B and 2 E (HEK293T, E-Cad and HTII-280) contain empty panels. We realise that these are meant to be control images, however, some level of background signal would be expected. Please provide source data for these panels.
14. In manuscript text file there are legends and callouts for tables and movies that have not been uploaded. Depending on the complexity, tables should be renamed to either Table EV1, etc. or Dataset EV1 etc. with legends included as separate tabs in each Excel file, or above tables if in .doc format. Videos should be renamed to Movie EV1, etc. and zipped individually in a zip folder with a corresponding legend (in word or README format). Further information is available here: <https://www.embopress.org/page/journal/14602075/authorguide#expandedview>
15. Our data editors have flagged the following issues in figure legends that need correcting:
 - Please define the annotated p values ***/**/* as well as provide the exact p-values for the same in the legend of figure EV 6c as appropriate.
 - Please provide the exact p values in the legends of figures 3h; EV 1b; EV 8a.

- Please indicate the statistical test used for data analysis in the legends of figures 4f, g, j; EV 4b-d", g.
- Please provide information on the number and nature of replicates in the legends of figures 4f, g, j.
- Please describe the nature (e.g., biological or technical) of replicates in the legend of figure EV 1b.
- Please define the error bars in the legends of figures 2k; 4f, g, j.
- Please provide a numbered scale bar for heatmap presented in figure EV 4j.
- Please define the scale bar for figures 1b-c; 2d, g; 4h, k; EV 2a-f; EV 7e, k.
- Please note that scale bar and its definition are missing for figure 2b.
- Please define the asterisk in the legend of figure EV 7g.
- Please define the cyan arrowheads/arrows in the legend of figures 2i-j.
- Please note that the scale bar for figure is mentioned as 1µm in the figure as opposed to 2µm in the legend of figure 1d. This needs to be rectified.

16. Papers published in The EMBO Journal are accompanied online by a 'Synopsis' to enhance discoverability of the manuscript. It consists of A) a short (1-2 sentences) summary of the findings and their significance, B) 3-4 bullet points highlighting key results and C) a synopsis image that is 550x300-600 pixels large (width x height, jpeg or png format). You can either show a model or key data in the synopsis image. Please note that the image size is rather small and that text needs to be readable at the final size. Please send us this information together with the revised manuscript.

With best wishes,

leva

 leva Gailite, PhD
 Senior Scientific Editor
 The EMBO Journal
 Meyerhofstrasse 1
 D-69117 Heidelberg
 Tel: +4962218891309
 i.gailite@embojournal.org

We realize that it is difficult to revise to a specific deadline. In the interest of protecting the conceptual advance provided by the work, we recommend a revision within 3 months (5th Jan 2025). Please discuss the revision progress ahead of this time with the editor if you require more time to complete the revisions.

 Referee #1:

Comment 1: This comment pertains to HTII-280 expression and levels in fetal AT2s in their organoids. As the authors justified, it is well known that HTII-280 is expression is very heterogenous in fetal tissues and it is expressed in all cells. Therefore, the flow plots presented in Ext Fig2G seems acceptable. And their use of negative control further supports the flow data.

Comment-2: In this comment, the reviewer points at the discrepancy between flow data, immunostaining images, and scRNA-seq data. I agree with the reviewer that there is some discrepancy between these data. However, as authors acknowledged, slight technical variations are very common in ex vivo cultured cells. Repeating these experiments is not only expensive but takes many months. In my view, this is not necessary for the main conclusions of this manuscript. The key findings have much more meaning than the slight heterogeneity observed in one data assay. In fact, the authors must be commended for including data from all modalities despite seeing some discrepancy between these assays. The future studies will figure out underlying causes.

Overall, this manuscript offers many insights into the mechanisms of surfactant biogenesis and its implications to interstitial lung diseases. Additionally, the authors have taken a technical tour-de-force to conduct some otherwise challenging experiments.

Comment-3: The authors provided revised data for this comment. This is acceptable.

Comment-4: The reviewer is asking for additional data to support the claim that the organoids can be passed for over 20 times. Again, this is not only expensive but takes many months. The data in Ext Data Fig-2 supports the authors claims. Additionally, the data from CRISPR based experiments indicates that the cells can be cultured for a long time. In my view, repeating these experiments is a waste of time and resources.

The authors addressed the remaining editorial issues.

Dear Emma,

Thank you for addressing the final editorial points. I am now pleased to inform you that your manuscript has been accepted for publication.

Before we forward your manuscript to the publishers, I would like to suggest minor edits in the manuscript abstract and synopsis. I have also written a blurb that will accompany the title of your manuscript on our website. Please take a look at the text below and in the attached file and let me know if any corrections are needed.

Blurb:

A forward genetic screen in human fetal lung-derived alveolar type 2 cell organoids identifies the E3 ligase ITCH as a regulator of correct surfactant protein C trafficking.

Synopsis

Dysfunction of alveolar type 2 cells and their surfactant secretion function has been linked to lung diseases in humans. This study describes the development of genetically manipulable, mature human fetal-derived alveolar type 2 cell organoids that facilitate the in vitro investigation of human disease genes.

- Cryopreservable alveolar type 2 cell organoids, which can self-renew and differentiate to alveolar type 1 cells, can be derived from human fetal lungs.
- Fetal lung-derived alveolar type 2 (fdAT2) organoids traffic and secrete surfactant appropriately and have similar gene expression profiles to adult alveolar type 2 cells.
- Knock-down of the E3 ligase ITCH in fdAT2 organoids phenocopies a pathological surfactant protein C trafficking defect seen in a subset of individuals with pulmonary fibrosis.

If you have any questions, please do not hesitate to contact the Editorial Office. Thank you for this contribution to The EMBO Journal and congratulations on a nice study!

Best wishes,

Ieva
